# A large-eddy simulation analysis of collective wind farm axial-induction set points in the presence of blockage

**Théo Delvaux and Johan Meyers**

Department of Mechanical Engineering, KU Leuven, Celestijnenlaan 300 – box 2421, 3001 Leuven, Belgium

**Correspondence:** Théo Delvaux (theo.delvaux@kuleuven.be)

**Abstract.** [TS1] Over the past few years, numerous studies have shown the detrimental impact of flow blockage on wind farm power production. In the present work, we investigate the benefits of a simple collective axial-induction set point strategy for power maximization and load reduction in the presence of blockage. To this end, we perform a series of large-eddy simulations (LESs) over a wind farm consisting of 100 IEA 15 MW turbines and build wind farm power and thrust coefficient curves under three different conventionally neutral boundary layers and one truly neutral boundary layer. As a result of the large-scale effects, we show that the wind farm power and thrust coefficient curves deviate significantly from those of an isolated turbine. We carry out a trade-off analysis and determine that, while the optimal thrust set point is still correctly predicted by the Betz limit under wake-only conditions, it shifts towards lower operating regimes under strong blockage conditions. In such cases, we observe a minor power increase with respect to the Betz thrust set point, accompanied by a load reduction of about 5 %. More interestingly, we show that for some conditions the loads can be reduced by up to 19 %, at the expense of a power decrease of only 1 %.

## 1 Introduction

Due to various constraints related to infrastructure costs, land regulations and grid connection, wind turbines are often gathered in wind farms. However, such configuration introduces non-negligible coupling between the turbines as upstream rows shed wakes on their downstream counterparts. This results in a large proportion of turbines in the farm facing lower incoming velocities and higher levels of turbulence intensity. Therefore, the design of an optimal wind farm operating strategy has been the focus of numerous research works (Steinbuch et al., 1988; Gebraad, 2014; González et al., 2015; Fleming et al., 2017; Annoni et al., 2017). To date, these strategies essentially consist of adjusting either the thrust coefficients (axial-induction control) or the yaw angles (wake redirection control) of the turbines in the farm.

Although many studies on optimal farm operating points have shown promising results, the majority builds upon low-fidelity engineering models, in which only the wake interactions are represented. However, recent research has highlighted the excitation of gravity waves by the farm on a much larger scale, with non-negligible impacts on the total power production of the farm (Allaerts and Meyers, 2018; Lanzilao and Meyers, 2022, 2024). This is associated with an unfavorable pressure gradient that is established at the inlet of the farm, leading to the so-called blockage effect.

To represent wind-farm-induced pressure effects on the upstream flow, Allaerts and Meyers (2019) developed an atmospheric perturbation model. With this model, they built a farm-averaged power coefficient curve for two sets of flow conditions. In both cases, they observed a significant drop with respect to the predictions of the wake-only models. In the work of Allaerts and Meyers (2019), only homogeneous distributions of the thrust coefficient were considered. Later, Lanzilao and Meyers (2021) proposed a more advanced optimization procedure of the wind farm thrust set point under specified flow conditions. The authors leveraged the analytical form of the model of Allaerts and Meyers (2019) to derive its adjoint gradient, with which they retrieved optimal thrust coefficient distributions over the farm. Overall, they observed power gains larger than 4 % in the majority of the

tested cases. More generally, their work emphasized the important part played by gravity-wave-induced blockage effects in the design of an optimal wind farm thrust set point. However, the approach proposed by Lanzilao and Meyers (2021) relied on a box-function wind farm force, with which the interactions between the turbines could not be accurately described. Using coupled wake-blockage models, Bossanyi and Bleeg (2024) recently pointed out that axial-induction control could reduce blockage and wake effects simultaneously.

In this context, the present work aims at providing solid evidence of the benefits that can possibly be achieved through collective wind farm axial-induction control of the thrust set point. For this purpose, we build the power coefficient and thrust coefficient curves of a large wind farm using high-fidelity large-eddy simulations (LESs). In this analysis, we investigate the impact of atmospheric conditions on the shape of the curves by considering four sets of flow conditions. Due to their high computational cost, LES data of full wind farm flows are scarce. Therefore, to the best of our knowledge, no similar study has been performed before.

The remainder of this paper is organized as follows. In Sect. 2, we introduce the set of governing equations, the boundary conditions, the numerical specifications and the different tested cases. Section 3 then provides details on the precursor and spin-up phases preceding the actual simulations. The results of the last simulations are discussed in Sect. 4, in terms of the flow fields and the wind farm performances.

## 2  Methodology

In this section, we introduce the equations governing the LESs, and we give a brief description of the flow solver used (Sect. 2.1). We then discuss the characteristics of the turbines and their representation in the numerical frame (Sect. 2.2). The boundary conditions selected in the scope of this study are described in Sect. 2.3, and further details about the numerical set-up are provided in Sect. 2.4. Eventually, the different atmospheric conditions and turbine thrust set points investigated in this work are summarized in Sect. 2.5. We emphasize that the methodology described below is, to a large extent, inspired by the one followed by Lanzilao and Meyers (2024).

### 2.1  Governing equations

Throughout the present work, the three-dimensional filtered velocity field is described by the incompressible Navier–Stokes equations. The Boussinesq approximation is used, and we employ a transport equation for the filtered potential temperature. The set of equations is explicitly given in Appendix A, and an in-depth description of the equations can be found in Allaerts and Meyers (2017).

Within this paper, we focus on barotropic flows, in which a constant background pressure gradient across the domain balances the Coriolis force above the capping inversion, resulting in a geostrophic wind in the free atmosphere that is constant with height. Moreover, the forcing exerted by the turbines on the flow (see Sect. 2.2) is accounted for through an actuator disk model (ADM). With regard to the subgrid-scale model, we use the stability-dependent Smagorinsky model developed by Stevens et al. (2000). The corresponding Smagorinsky coefficient is set to $C_s = 0.14$, similar to previous works carried out with SP-Wind (Goit and Meyers, 2015; Allaerts and Meyers, 2017; Munters and Meyers, 2018; Lanzilao and Meyers, 2023, 2024). Moreover, we use the damping approach of Mason and Thomson (1992) near the wall, which is a well-established technique for neutral atmospheric boundary layers (ABLs) (see also Meyers, 2011).

In order to solve the set of equations, we use the in-house SP-Wind solver (see, e.g., Calaf et al., 2010; Goit and Meyers, 2015; Allaerts and Meyers, 2017, 2018; Munters and Meyers, 2018; Lanzilao and Meyers, 2022; Lanzilao and Meyers, 2023). This software relies on a classical fourth-order Runge–Kutta scheme with a Courant–Friedrichs–Lewy number of 0.4 to integrate the system in time. At every stage of the numerical scheme, the Poisson equation is solved to ensure continuity. Further, SP-Wind provides pseudo-spectral Fourier schemes that are used to discretize the equations along the streamwise and spanwise directions. We note that aliasing errors are prevented thanks to the 3/2 dealiasing rule of Canuto et al. (1998). Finally, the vertical direction is discretized following an energy-preserving fourth-order finite-difference scheme, as discussed in Verstappen and Veldman (2003). The reader can refer to, e.g., Delport (2010) for further details about the discretization employed.

### 2.2  Wind turbine characteristics

In this paper, we model the performances of the IEA 15 MW offshore turbine detailed by Gaertner et al. (2020). It is equipped with a rotor diameter $D = 240$ m located at hub height $z_H = 150$ m that delivers a rated power of $P_r = 15$ MW.

We model the turbine rotor as a non-rotating actuator disk, similar to the LES studies of Allaerts and Meyers (2015), Goit and Meyers (2015), and Lanzilao and Meyers (2022, 2023, 2024), among others. In the actuator disk model, the turbine acts as an infinitely thin disk that extracts momentum from the flow. However, in order to prevent numerical instabilities associated with abrupt gradients of forces, we smooth out the force distribution by means of a Gaussian filtering operation (Calaf et al., 2010; Meyers and Meneveau, 2010). We define the three-dimensional Gaussian filter as

$$G(\boldsymbol{x}) = \left( \frac{6}{\pi \Delta_f^2} \right)^{3/2} \exp\left[ -6 \frac{x^2 + y^2 + z^2}{\Delta_f^2} \right], \qquad (1)$$

where $\boldsymbol{x}$ denotes the coordinate vector, and $\Delta_f$ is the filter width. In SP-Wind, the filter width relates to the grid spac-

ing through $\Delta_f = \max\{f_x \Delta x, f_y \Delta y, f_z \Delta z\}$, with $f_x = f_y = f_z = 1.5$ being the filter parameters and $\Delta x$, $\Delta y$ and $\Delta z$ being the cell dimensions discussed in Sect. 2.4. Consequently, the footprint for a turbine centered at $\boldsymbol{x}_t$ corresponds to (Meyers and Meneveau, 2010)

$$
\mathcal{R}(\boldsymbol{x}) = \int \int \int_{\Omega} G(\boldsymbol{x} - \boldsymbol{x}') \delta \left[ (\boldsymbol{x}' - \boldsymbol{x}_t) \cdot \boldsymbol{e}_{\perp} \right]
$$
$$
\times \mathcal{H}(D/2 - ||\boldsymbol{x}' - \boldsymbol{x}_t||_2) \mathrm{d}\boldsymbol{x}', \tag{2}
$$

with $\boldsymbol{e}_{\perp}$ being the unit vector orthogonal to the turbine and $\Omega$ being the three-dimensional space. In Eq. (2), the symbols $\delta$ and $\mathcal{H}$ represent the Dirac delta distribution and the Heaviside function, respectively.

Then, the velocity at the location of the $k$th rotor and perpendicular to it is computed as the spatial average over the footprint:

$$
u_{d,k} = \frac{M}{A} \int \int \int_{\Omega} \mathcal{R}(\boldsymbol{x}) \boldsymbol{u} \cdot \boldsymbol{e}_{\perp} \mathrm{d}\boldsymbol{x}, \tag{3}
$$

where $A = \pi D^2 / 4$, and $M$ is the velocity correction factor (Shapiro et al., 2019). For coarse grids, the filtering operation may lead to a power overestimation as the rotor diameter appears to be artificially increased. Therefore, we use the velocity correction factor $M$ proposed by Shapiro et al. (2019) as a function of the filter width $\Delta_f$ and the disk-based thrust coefficient $C_T'$:

$$
M = \left( 1 + \frac{C_T'}{2} \frac{1}{\sqrt{3\pi}} \frac{\Delta_f}{D} \right)^{-1}. \tag{4}
$$

Furthermore, the magnitude of the thrust force exerted by the $k$th rotor,

$$
F_k = \frac{1}{2} \rho_0 C_{T,k}' u_{d,k}^2 \frac{\pi}{4} D^2, \tag{5}
$$

is distributed over the turbine footprint as done by Meyers and Meneveau (2010),

$$
f_k(\boldsymbol{x}) = F_k \mathcal{R}(\boldsymbol{x}), \tag{6}
$$

where $\rho_0$ is the reference air density and $u_{d,k}$ the local disk-averaged velocity (Eq. 3). Further, the force intensity is set through the disk-based thrust coefficient denoted $C_T'$, the value of which is given as input to SP-Wind (see Sect. 2.5). Similarly to Allaerts and Meyers (2017) and Lanzilao and Meyers (2024), we employ a simple yaw controller that maintains the actuator disk perpendicular to the flow direction measured 1 D upstream. Consequently, we emphasize that the total force $F_k$ (Eq. 5) is the magnitude of a vector that generally has components along both the spanwise and the streamwise directions. Finally, the total power the $k$th turbine extracts from the flow, denoted $P_k$, is computed as follows:

$$
P_k = F_k u_{d,k} = \frac{1}{2} \rho_0 C_{T,k}' u_{d,k}^3 \frac{\pi}{4} D^2. \tag{7}
$$

## 2.3 Boundary conditions

The boundary conditions of the numerical domain are specified as follows. On the bottom face, we model the development of shear stresses by means of the classic Monin–Obukhov similarity theory for a neutral boundary layer (Moeng, 1984; Allaerts, 2016), for which a surface roughness $z_0 = 1 \times 10^{-4}$ m, representative of offshore conditions, is selected.

Further, both the streamwise and the spanwise lateral sides of the domain are assigned periodic boundary conditions. This allows us to model an infinitely wide domain, provided that no farm-induced effects reach the edge of the domain. The choice of an appropriate domain size is discussed in Sect. 2.4. Along the streamwise direction, we employ the wave-free fringe region technique developed by Lanzilao and Meyers (2023), in which a body force is applied to ensure that the desired inflow conditions are imposed at the front of the domain. The generation of spurious gravity waves arising from this non-physical body force is prevented thanks to a damping of the vertical momentum above the ABL. The wave-free fringe region technique is used together with a concurrent precursor approach, from which the fully developed turbulent flow field can be imposed (see Sect. 3.1).

At the top of the domain, a rigid-lid condition ensures zero shear stress, zero vertical velocity and a fixed potential temperature. Without particular treatment, however, this boundary condition significantly reflects gravity waves. Therefore, we use a Rayleigh damping layer (RDL) to curtail this wave reflection effect, similar to Allaerts and Meyers (2017) and Lanzilao and Meyers (2023, 2024), among others. The method consists of applying a body force in the upper part of the free atmosphere, with an intensity proportional to the difference between the local velocity field and the geostrophic wind.

## 2.4 Numerical set-up

Prior to simulating the flow in the wind farm, we run a precursor simulation in which the turbulent flow fully develops and reaches a statistically steady behavior. When running the wind farm simulation, the precursor is concurrently advanced in time so as to provide the inflow, as discussed in Sect. 2.3. In the scope of this work, we select a precursor domain with dimensions $L_x^p = L_y^p = 10$ km and $L_z^p = 3$ km, as done by Allaerts and Meyers (2017, 2018) and Lanzilao and Meyers (2024). Next, we set the dimensions of the main domain on the basis of the observations of Lanzilao and Meyers (2024). We note that in the latter study, the authors consider a wind farm about 4 km longer but 2 km narrower than the one investigated in the present work (see Sect. 2.5). Therefore, we use the same main domain length and height as in Lanzilao and Meyers (2024), i.e., $L_x \times L_z = 50$ km $\times$ 25 km, but we increase the domain width by 10 km so that the main domain has dimensions

$L_x \times L_y \times L_z = 50\,\mathrm{km} \times 40\,\mathrm{km} \times 25\,\mathrm{km}$. While the domain height may initially appear overly large, it is required to allow for the non-reflecting radiation of gravity waves and to accommodate the Rayleigh damping layer described in Sect. 2.3. The farm is symmetrically positioned along the spanwise direction, resulting in a distance of $L_{\mathrm{side}} = 14.3\,\mathrm{km}$ between the edges of the farm and the lateral sides of the domain. Eventually, the distance upstream of the farm is taken equal to $L_{\mathrm{ind}} = 18\,\mathrm{km}$ to allow for a full representation of the induction zone (Lanzilao and Meyers, 2024).

Because periodicity is imposed over the four lateral sides of the precursor domain, the tiling technique of Sanchez Gomez et al. (2023) is employed to extend the 10 km long and 10 km wide precursor field to the horizontal dimensions of the main domain. The resulting field is used as the initial state in the wind farm simulations. The same tiling operation, limited to the spanwise direction however, is carried out to generate the concurrent precursor with horizontal dimensions $L_x^{\mathrm{cp}} \times L_y^{\mathrm{cp}} = 10\,\mathrm{km} \times 40\,\mathrm{km}$. Additionally, we artificially extend the height of the precursor field by imposing the geostrophic flow field from 3 to 25 km for all the considered atmospheric conditions. The characteristics of the precursor field are further discussed in Sect. 3.1.

Furthermore, the grid resolution is identical to that selected by Lanzilao and Meyers (2024), that is, $\Delta x = 31.25\,\mathrm{m}$ and $\Delta y = 21.74\,\mathrm{m}$ along the streamwise and spanwise directions, respectively. This corresponds to $N_x^{\mathrm{p}} = 320$ and $N_y^{\mathrm{p}} = 460$ grid points along the $x$ and $y$ axes of the precursor domain. In the main domain, the selected resolution leads to $N_x = 1600$ and $N_y = 1840$ points. Contrary to the regular grid spacing adopted in the horizontal plane, we use a height-dependent vertical discretization to reduce the computational cost. Firstly, a relatively fine constant spacing $\Delta z = 5\,\mathrm{m}$ is retained below 1.5 km, with which the velocity gradients can be accurately captured. Consequently, the turbine rotor encompasses 11 and 48 grid points along the spanwise and vertical directions, respectively. We note that these values align with those of other recent similar studies (Calaf et al., 2010; Wu and Porté-Agel, 2011; Allaerts and Meyers, 2017; Lanzilao and Meyers, 2024). Secondly, the grid is smoothly stretched over 180 points in the region between 1.5 and 15 km. Lastly, an additional stretch is applied over 10 grid points from 15 to 25 km. Overall, a total of $N_z = 490$ grid points are used along the vertical direction. Note that the same vertical discretization, trimmed to $L_z^{\mathrm{p}} = 3\,\mathrm{km}$, however, is adopted for the initial precursor simulation. In the region where the vertical grid spacing is the finest, the spanwise/vertical aspect ratio is equal to $\Delta y / \Delta z \simeq 4.3$. Although no detailed study on the aspect ratio impact has been performed with SP-Wind, values of the order of 3 to 4 have historically been retained (Allaerts and Meyers, 2017; Lanzilao and Meyers, 2023; Lanzilao and Meyers, 2024) in order to account for the differences between the discretization schemes used in the spanwise and vertical directions (see Sect. 2.1).

**Table 1.** Magnitude ($\nu^{\mathrm{ra}}$), growing rate ($s^{\mathrm{ra}}$) and thickness ($L_z^{\mathrm{ra}}$) of the Rayleigh damping layer. Parameter values are set following Lanzilao and Meyers (2024).

| Parameter | $\nu^{\mathrm{ra}}$ | $s^{\mathrm{ra}}$ | $L_z^{\mathrm{ra}}$ |
|---|---|---|---|
| Value | 5.15 | 3 | 10 |
| Unit | – | – | km |

In Sect. 4.2, the power production of an isolated turbine is compared to that of the wind farm for reference. Therefore, it is necessary to perform simulations of an identical turbine operating in standalone conditions. The horizontal dimensions of the corresponding main domain are the same as those of the precursor simulation so that only the vertical extension from 3 to 25 km is required.

Finally, as the wind farm set-up and the domain size considered here are very similar to those of Lanzilao and Meyers (2024), the same settings are selected for the Rayleigh damping layer and for the fringe region. The corresponding values are summarized in Tables 1 and 2, respectively. In Table 1, the first two parameters denote the magnitude and the growing rate of the RDL, whereas $L_z^{\mathrm{ra}}$ is the thickness of the layer. The first four parameters in Table 2 refer to the starting and ending points of the fringe region and the corresponding smoothness coefficients. The following four parameters denote the same quantities but related to the vertical-momentum-damped region. Eventually, $h_{\mathrm{max}}$ characterizes the strength of the fringe function. The mathematical expressions of the Rayleigh damping function, the fringe region forcing and the momentum damping function are given in Appendix B.

## 2.5    Wind farm operating conditions

The wind farm examined in the current work consists of 100 IEA 15 MW turbines arranged in a 10-by-10 configuration. The spacing between each turbine is set to $S_x = S_y = 5\,D$ in both the spanwise and the streamwise directions. We introduce an offset of $S_y/2$ between every downstream row to obtain a staggered layout. Given the turbine diameter specified in Sect. 2.2, the resulting power density is $P_{\mathrm{r}}/(S_x S_y) \simeq 10.42\,\mathrm{MW\,km^{-2}}$. Overall, the wind farm, starting at $L_{\mathrm{ind}} = 18\,\mathrm{km}$ (Sect. 2.4), is $L_x^{\mathrm{f}} = 10.8\,\mathrm{km}$ long and $L_y^{\mathrm{f}} = 11.4\,\mathrm{km}$ wide, leading to the following ratios: $L_{\mathrm{ind}}/L_x^{\mathrm{f}} = 1.67$, $L_x/L_x^{\mathrm{f}} = 4.63$ and $L_y/L_y^{\mathrm{f}} = 3.51$. A sketch of the numerical domain is depicted in Fig. 1.

In order to explore the potential for power optimization and load reduction using axial-induction control, different disk-based thrust coefficients (Eq. 5) are tested. From axial momentum theory, the value $C_{\mathrm{T}} = 8/9$, or equivalently $C_{\mathrm{T}}' = 2$ (Allaerts, 2016), maximizes power extraction. In practice, the designed thrust set point is slightly lower to reduce the associated loads at rated wind speed (Gaertner et al., 2020).

**Table 2.** Starting $(x_s^h)$ and ending $(x_e^h)$ points of the fringe region and corresponding smoothness coefficients $(\delta_s^h, \delta_e^h)$. Starting $(x_s^d)$ and ending $(x_e^d)$ points of the momentum-damped region and corresponding smoothness coefficients $(\delta_s^d, \delta_e^d)$. Fringe region strength $(h_{max})$. Parameter values are selected following Lanzilao and Meyers (2024).

| $x_s^h$ | $x_e^h$ | $\delta_s^h$ | $\delta_e^h$ | $x_s^d$ | $x_e^d$ | $\delta_s^d$ | $\delta_e^d$ | $h_{max}$ |
|---|---|---|---|---|---|---|---|---|
| 44.5 km | 47.2 km | 0.4 km | 0.4 km | 44.5 km | 50 km | 2.5 km | 3 km | 0.3 s$^{-1}$ |

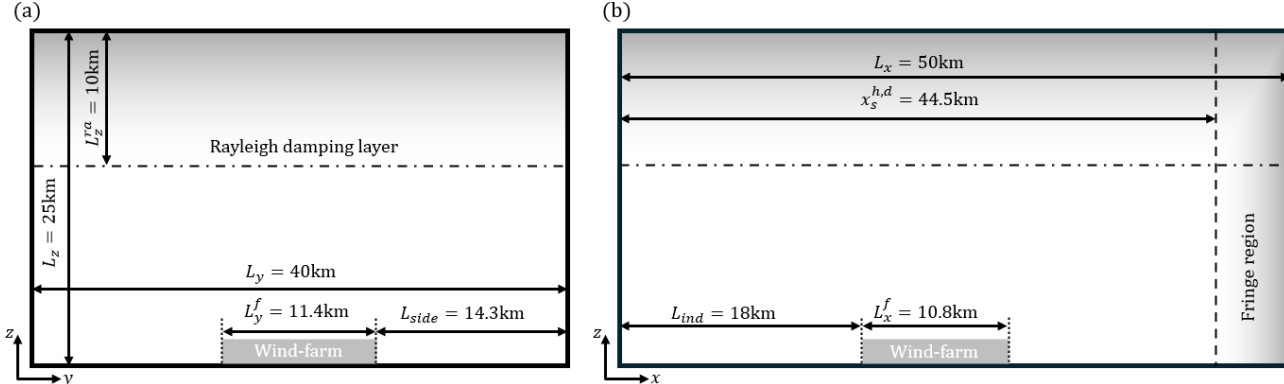

**Figure 1.** Scaled representation of the front **(a)** and side **(b)** views of the domain set-up employed in the wind farm simulations. The Rayleigh damping layer and the fringe region introduced in Sect. 2.4 are shown in the figure.

In the scope of this work, we therefore select the theoretical optimal disk-based thrust coefficient, along with three other values of $C_T'$ evenly spaced at intervals of 0.75: $C_T' = \{0.50, 1.25, 2.0, 2.75\}$. Following classical momentum theory (Allaerts, 2016), the corresponding values of the thrust coefficient are $C_T = \{0.40, 0.73, 0.89, 0.97\}$. We emphasize that in all the simulations, the considered $C_T'$ value is constant throughout the wind plant, representing the choice of a collective thrust set point. This allows us to restrict the number of parameters in the study.

To initialize the precursor simulation, a potential temperature profile is defined following the model of Rampanelli and Zardi (2004) for conventionally neutral boundary layers (CNBLs). We denote the height of the capping inversion by $H$ and set the constant potential temperature below it to $\theta_0 = 288.15$ K. Further, we refer to the strength and thickness of the capping inversion as $\Delta\theta$ and $\Delta H$, respectively. Above the capping inversion, the potential temperature profile is controlled by the rate $\Gamma$ in the free atmosphere.

Based on the observations of Lanzilao and Meyers (2024), we select a first set of parameters, $\{H = 150\,\text{m}, \Delta\theta = 8\,\text{K}, \Gamma = 1\,\text{K km}^{-1}\}$ (referred to as $H150$–$\Delta\theta8$–$\Gamma1$), for which strong blockage effects are expected to have a substantial influence on the wind farm efficiency. Lanzilao and Meyers (2024) reported a non-local efficiency lower than 0.3, however partially counterbalanced by a strong favorable pressure gradient within the farm, with which the wake efficiency becomes larger than 1. In spite of this, the $H150$–$\Delta\theta8$–$\Gamma1$ case was observed to result in a low farm efficiency of about 32 %. Secondly, we consider the scenario $H300$–$\Delta\theta5$–$\Gamma1$, in which the blockage effect is attenuated due to a weaker capping inversion positioned at a higher altitude. Finally, we investigate the combination $H500$–$\Delta\theta5$–$\Gamma4$, for which Lanzilao and Meyers (2024) observed a beneficial impact of the thermal stratification on the farm efficiency. Note that for all three sets of atmospheric conditions, the capping inversion thickness is initialized to $\Delta H = 100$ m. In addition to the three CNBL atmospheric conditions, we consider a situation with no thermal stratification, similar to Lanzilao and Meyers (2024). To generate this flow, we start from the $H500$–$\Delta\theta5$–$\Gamma4$ case and artificially set a constant potential temperature profile when copying the solution from the precursor to the main domain. The resulting flow, denoted $H500$–$\Delta\theta0$–$\Gamma0$, resembles a truly neutral boundary layer (TNBL) but with the same inlet velocity as $H500$–$\Delta\theta5$–$\Gamma4$.

Moreover, for all the simulations performed in this analysis, we set the geostrophic wind speed to $G = 10\,\text{m s}^{-1}$, as done by Lanzilao and Meyers (2024). We remark that this speed is slightly lower than the rated speed of the IEA 15 MW turbine reported by Gaertner et al. (2020). Consequently, this choice of geostrophic speed allows us to analyze the turbine performances in the region where it typically operates at maximum thrust coefficient when following a greedy control approach. Finally, we set a latitude $\phi = 51.6°$, resulting in a Coriolis frequency $f_c = 1.14 \times 10^{-4}\,\text{s}^{-1}$.

## 3 Boundary-layer initialization

The precursor phase performed to initialize the boundary-layer flow is described in Sect. 3.1. Then, the wind farm set-up introduced in Sect. 2.5 is added to the main domain, and a spin-up phase is conducted until the flow reaches a quasi-steady state. The transient behavior of the flow during this second phase is discussed in Sect. 3.2.

### 3.1 Precursor phase

The precursor phase is carried out to obtain a statistically steady, fully developed turbulent flow over the domain. For this purpose, the initial velocity profiles are defined following the approach of Allaerts and Meyers (2015), that is, a boundary-layer flow with friction velocity $u_* = 0.26 \, \mathrm{m \, s^{-1}}$ connected to a laminar geostrophic wind above the capping inversion. Turbulence is initiated by means of divergence-free fluctuations of amplitude $G/10$ introduced up to an altitude of 100 m. The initial potential temperature profiles are generated using the Rampanelli and Zardi (2004) model together with the sets of parameters, $H$, $\Delta\theta$ and $\Gamma$, detailed in Sect. 2.5. We emphasize that a Rayleigh damping layer is also applied during the precursor phase to damp the inertial fluctuations and the gravity waves above 1 km in the atmosphere. For each atmospheric condition, the precursor simulation is performed over 20 h. The resulting flow quantities are then time averaged over the last 4 h of the simulation and are displayed in Fig. 2.

Figure 2 shows the velocity (Fig. 2a) and the potential temperature profiles (Fig. 2d) averaged over the horizontal planes, together with the corresponding shear stress profiles (Fig. 2b) and wind directions (Fig. 2c). From Fig. 2a, it can be seen that the presence of the capping inversion limits the boundary-layer growth so that the equilibrium inversion-layer height is attained when buoyancy forces balance the surface shear stress (Csanady, 1974). As observed by Lanzilao and Meyers (2024), the amplitude of the super-geostrophic jet that forms at the top of the ABL increases with decreasing inversion-layer heights. Above the jet, the shear stress profile reduces to zero (Fig. 2b), the flow becomes laminar and the velocity profile corresponds to the geostrophic wind. Figure 2d shows that for the cases $H150$–$\Delta\theta8$–$\Gamma1$, $H300$–$\Delta\theta5$–$\Gamma1$ and $H500$–$\Delta\theta5$–$\Gamma4$, the origin of the capping inversion moves to an altitude of 195, 325 and 510 m, respectively, over the 20 h long spin-up. We note that these values align with the predictions computed from Csanady (1974) (not detailed here). In the ABL, the Ekman spiral forms so that the wind direction angle $\Phi_{\mathrm{d}}$, measured with respect to the axis perpendicular to the farm, varies with the altitude (Fig. 2c). The angle $\Phi_{\mathrm{d}}$ values in Fig. 2c are normalized by the largest value of $|\alpha|$, where $\alpha$ is defined as the angle between the geostrophic wind and the velocity vector right above the ground. As reported by Allaerts and Meyers (2017) and Lanzilao and Meyers (2024), $|\alpha|$ is ob-

served to be larger for lower capping inversions. Note that the wind direction controller designed by Allaerts and Meyers (2015) is employed during the precursor phase to rotate the geostrophic wind so as to ensure there are no spanwise velocity components at hub height, i.e., $\Phi(z_{\mathrm{hub}}) = 0°$. Finally, small oscillations of the velocity magnitude and the wind direction appear in the inversion layer (Fig. 2a and b) as a result of the strong stratification that characterizes this region. This matter is addressed in Sullivan et al. (2016), where the authors show that eddies with a characteristic scale larger than the Dougherty–Ozmidov length are stratification dependent. However, this length decreases as stratification increases, possibly leading to values of the Dougherty–Ozmidov length that are smaller than the grid spacing. Some of the subgrid-scale eddies generated in the inversion layer can be stratification dependent and can therefore not be accurately captured by the subgrid-scale (SGS) model, causing the oscillations observed in Fig. 2a and b. Similar oscillations can be seen in, e.g., Maas and Raasch (2022) and Pedersen et al. (2014).

### 3.2 Wind farm spin-up phase

The flow field generated at the last time step of the precursor phase is tiled over the concurrent precursor domain and the main domain described in Sect. 2.4. Then, we place the wind farm introduced in Sect. 2.5 in the main domain, and we advance the simulation in time so that the flow adapts to the presence of the farm. Simultaneously, the concurrent precursor flow evolves and is imposed in the fringe region following the methodology detailed in Sect. 2.3.

In the current work, we focus on obtaining accurate power estimations for a limited number of test cases. Therefore, we first check the convergence of the farm power. In Fig. 3, the evolution of the instantaneous wind farm power calculated over the two simulation phases is represented for all the considered conditions. We show the normalized difference with respect to the time-averaged power $\overline{P}_{\mathrm{a}}$ obtained in the second phase only, i.e., the last 60 min depicted in Fig. 3. From this same phase, we retrieve the standard deviation of $P_{\mathrm{a}}$ for each set of atmospheric conditions and operating conditions. This quantity is denoted $\sigma_{\mathrm{a}}$ and is represented in Fig. 3 to assess convergence. For all the considered cases, the normalized deviation $\sigma_{\mathrm{a}}/\overline{P}_{\mathrm{a}}$ is of the order of $10^{-2}$ and appears to increase slightly with $C'_{\mathrm{T}}$.

Figure 3 shows that, beyond 90 min, any remaining trend appears to be of the order of the power fluctuations, for all the operating regimes and the atmospheric conditions. Interestingly, we note that the statistically steady state is attained more rapidly for low-blockage conditions. Nevertheless, we retain a spin-up duration of 90 min, after which the final phase of the simulation is performed over 1 h. The power and flow quantities are measured during this last phase, referred to as the actual simulation. Similar to the precursor phase (Sect. 3.1), the wind direction controller of Allaerts and Meyers (2015) is employed during the wind farm spin-up phase.

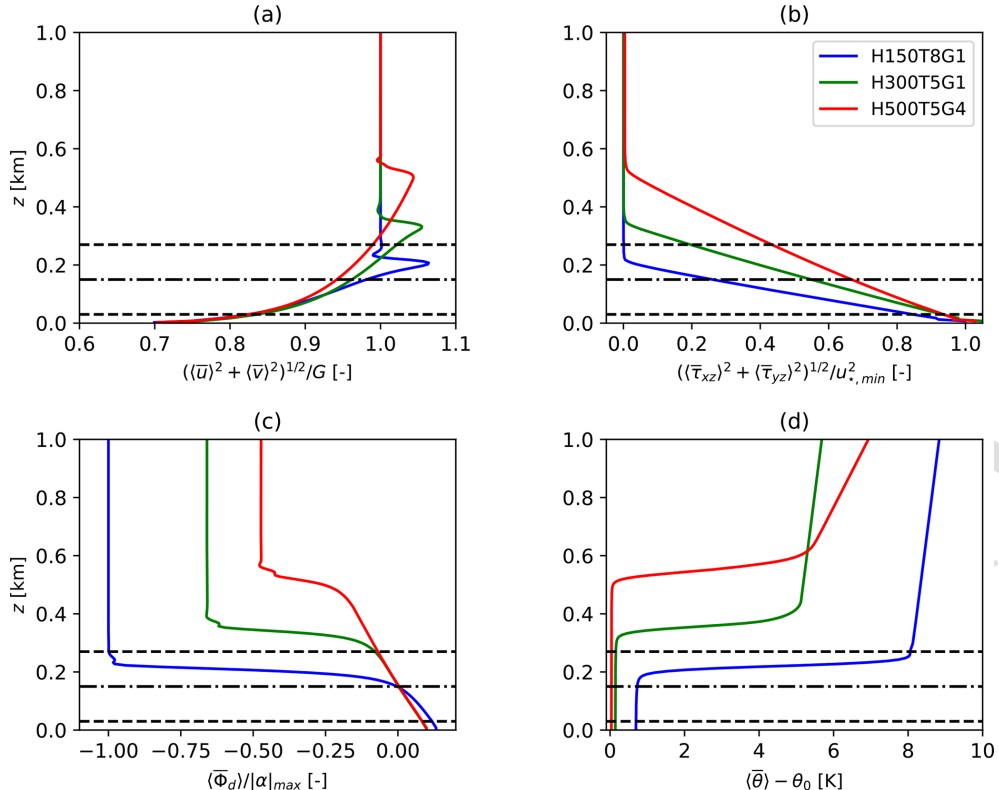

**Figure 2.** Vertical profiles of the velocity magnitude **(a)**, the total shear stress **(b)**, the wind direction **(c)** and the potential temperature **(d)**. The space-averaged profiles are computed over the last 4 h of the simulations for the three sets of atmospheric conditions and normalized by $G = 10 \, \mathrm{m \, s^{-1}}$, $u_{\star,\mathrm{min}} = 0.276 \, \mathrm{m \, s^{-1}}$, $|\alpha|_{\mathrm{max}} = 18.55°$ and $\theta_0 = 288.15 \, \mathrm{K}$, respectively. For all quantities, the top bar and the angle brackets represent time and horizontal averages, respectively. TS2

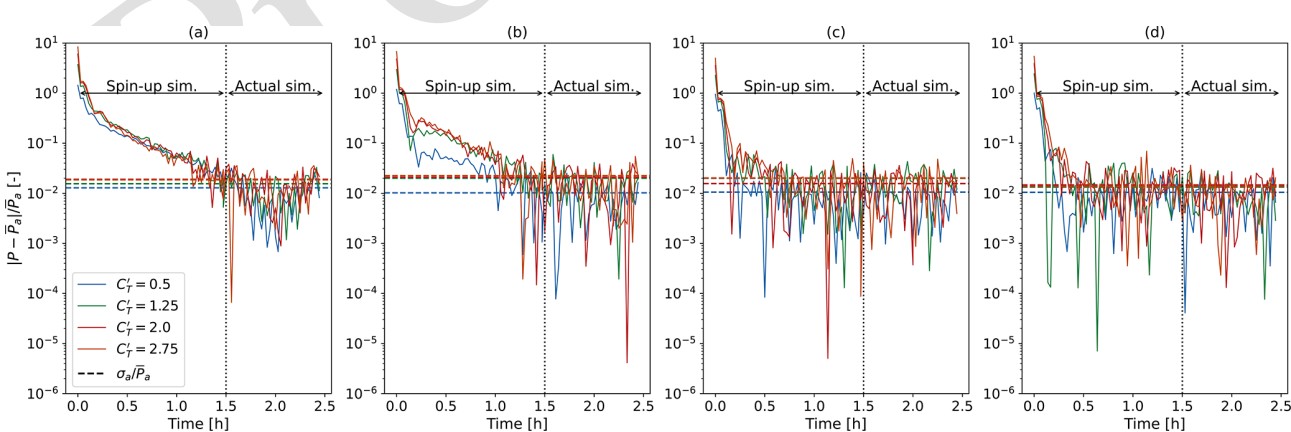

**Figure 3.** Evolution of the normalized power difference $|P - \overline{P}_a|$ measured during the two simulation phases for the four tested operating regimes and the four atmospheric conditions. Panels **(a)**–**(d)** correspond to conditions $H150$–$\Delta\theta 8$–$\Gamma 1$, $H300$–$\Delta\theta 5$–$\Gamma 1$, $H500$–$\Delta\theta 5$–$\Gamma 4$ and $H500$–$\Delta\theta 0$–$\Gamma 0$, respectively. For each of the 16 considered cases, the threshold defined by $\sigma_a / \overline{P}_a$ is represented by the dotted line of the corresponding color. The quantities $\overline{P}_a$ and $\sigma_a$ are the time-averaged power and corresponding standard deviation measured during the second phase only. TS3

This controller is disabled in the actual simulation, however. Eventually, the same procedure is applied to the corresponding single-wind-turbine cases over the small domain, as discussed in Sect. 2.4.

# 4   Results

The results of the 1 h long actual simulations are discussed in this section. First, we provide insights into the velocity fields in Sect. 4.1, pointing out the importance of the farm-induced effects. The corresponding power estimations are then analyzed in Sect. 4.2 to assess the potential of the collective axial-induction operational strategy.

## 4.1   Comparison of the farm-scale velocity fields

The instantaneous streamwise velocity field is provided in Fig. 4 for the four tested atmospheric conditions and the four operating regimes introduced in Sect. 2.5. Comparing the cases with an identical disk-based thrust coefficient ($C_T'$), the flow appears to vary significantly with the atmospheric conditions. This observation stresses the need to account for the potential temperature profile in the design of an efficient large-scale operating strategy. In particular, a large blockage effect is visible in the form of a bow wave in Fig. 4a–d. Even though this effect was anticipated due to the low and strong capping inversion above the farm, we observe a significant decrease in amplitude of this feature when $C_T'$ is decreased. The same pattern can be seen, though to a lesser extent, in the $H300–\Delta\theta5–\Gamma1$ case in Fig. 4e–h. Under the set of conditions $H500–\Delta\theta5–\Gamma4$ (Fig. 4i–l), large-scale effects are minor but become apparent when compared to the $H500–\Delta\theta0–\Gamma0$ case (Fig. 4m–p). In particular, analyzing Fig. 4l and p together, we observe a slight velocity decrease limited to the front of the farm in Fig. 4l. Moreover, Fig. 4p shows a stronger farm wake compared to Fig. 4l, providing evidence that a favorable pressure gradient still forms in situations where the capping inversion is high.

Eventually, observations of the vertical velocity field provided further evidence of the development of wind-farm-induced effects (not shown). In all the considered cases, the displacement of the capping inversion was seen to trigger internal gravity waves, yet to different degrees depending on the value of $C_T'$. Thus, stronger and weaker waves were observed for the cases where the farm operates at high and low $C_T'$ values, respectively. We refer to Lanzilao and Meyers (2024) for a more detailed analysis of this phenomenon.

## 4.2   Momentum extraction distribution across the farm

In order to assess the intensity of the thrust force exerted by the $k$th turbine on the flow and its corresponding power, we define the time-averaged thrust and power coefficients (denoted $C_{T,k}$ and $C_{P,k}$, respectively) as follows:

$$C_{T,k} = \frac{\overline{F}_k}{\frac{1}{2}\rho_0 A \overline{U}_\infty^2} \text{ and } C_{P,k} = \frac{\overline{P}_k}{\frac{1}{2}\rho_0 A \overline{U}_\infty^3}. \tag{8}$$

In these expressions, $A = \pi D^2/4$ is the disk area, and $\overline{F}_k$ and $\overline{P}_k$ are the time-averaged turbine thrust (Eq. 5) and power (Eq. 7), respectively. Further, $U_\infty$ is the reference wind speed computed as the streamwise velocity averaged over a layer of thickness $D$ spanning the disk-precursor domain, i.e., the region defined by $[0, L_x^p] \times [0, L_y^p] \times [z_H - D/2, z_H + D/2]$. Within this region, we use a vertically dependent weighted average where the weights are given by the actuator disk chord length, i.e., the straight-line distance across the intersection of the disk and the horizontal plane at the considered altitude. For the cases $H150–\Delta\theta8–\Gamma1$, $H300–\Delta5–\Gamma1$ TS4, $H500–\Delta\theta5–\Gamma4$ and $H500–\Delta\theta0–\Gamma0$, the reference speeds averaged over the last 4 h of the precursor simulation are equal to $\overline{U}_\infty = 9.61$, 9.55, 9.35 and 9.35 m s$^{-1}$, respectively. We note that the two expressions in Eq. (8) can be re-written as $C_{T,k} = C_T'(\overline{u}_{d,k}/\overline{U}_\infty)^2$ and $C_{P,k} = C_T'(\overline{u}_{d,k}/\overline{U}_\infty)^3$ using Eqs. (5) and (7), with $\overline{u}_{d,k}$ being the time average of $u_{d,k}$. The time-averaged thrust ($C_{T,sgl}$) and power ($C_{P,sgl}$) coefficients in the single-turbine case are defined with respect to the corresponding thrust ($\overline{F}_{sgl}$) and power ($\overline{P}_{sgl}$), analogously to Eq. (8).

The distribution of the local thrust coefficient ($C_{T,k}$) over the farm is normalized by that of the single turbine ($C_{T,sgl}$) under the same operating conditions and is represented in Fig. 5. Therefore, Fig. 5 illustrates the momentum extracted by each turbine in the farm compared to that of an isolated turbine. Because the disk-based thrust coefficient is common to each turbine in the farm, the ratio shown in Fig. 5 re-writes $C_{T,k}/C_{T,sgl} = \overline{F}_k/\overline{F}_{sgl} = (\overline{u}_{d,k}/\overline{u}_{d,sgl})^2$. When operating at $C_T' = 0.5$ (Fig. 5a, e, i and m), wake interference between the turbines dominates, which results in a region of higher thrust values over the first two rows of turbines, followed by a quasi-uniform distribution across the rest of the farm. In the absence of a capping inversion, the same conclusion applies regardless of the considered $C_T'$ value (Fig. 5m–p). For the CNBL conditions, the bow-wave pattern described in Fig. 4 is associated with a favorable pressure gradient that is, for example, visualized through thrust coefficients greater at the fourth row than at the third row in the $H150–\Delta\theta8–\Gamma1$ case (Fig. 5a). As $C_T'$ increases, the velocity at the farm entry decreases so that the row of minimal thrust coefficient is shifted towards the front of the farm (Fig. 4d). Interestingly, Fig. 5i–l show that for a high capping inversion, blockage essentially affects the first two rows. Consequently, it is possible to select the value of $C_T'$ common to all turbines so that the front-localized blockage and the wake effects downstream lead to a close-to-uniform thrust distribution across the farm (Fig. 5l). More generally, provided that the operating regime can be set independently for each turbine row, Fig. 5 indicates that the thrust distribution could be homogenized by either in-

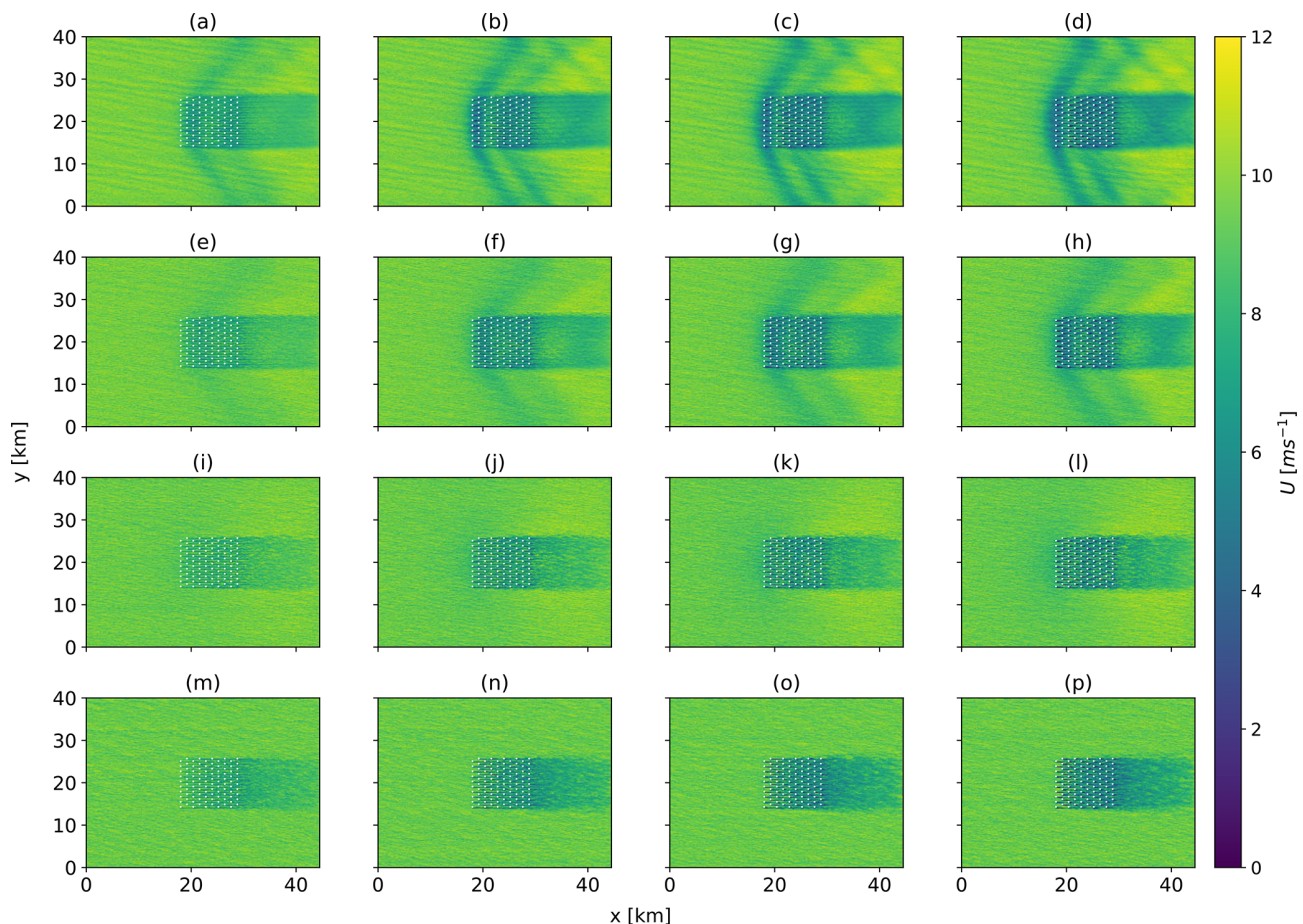

**Figure 4.** Instantaneous streamwise velocity field at hub height for the $H150$–$\Delta\theta8$–$\Gamma1$ **(a–d)**, $H300$–$\Delta\theta5$–$\Gamma1$ **(e–h)**, $H500$–$\Delta\theta5$–$\Gamma4$ **(i–l)** and $H500$–$\Delta\theta0$–$\Gamma0$ **(m–p)** cases. Four operating regimes are considered: $C'_T = 0.50$ **(a, e, i)**, $C'_T = 1.25$ **(b, f, j)**, $C'_T = 2.0$ **(c, g, k)** and $C'_T = 2.75$ **(d, h, l)**. The white markers indicate the turbine locations.

creasing or decreasing $C'_T$ at the front of the farm subject to high-blockage (e.g., Fig. 5d) or low-blockage conditions (e.g., Fig. 5i), respectively.

### 4.3 Wind farm thrust and power coefficient curves

We define the wind farm thrust and power coefficients, denoted $C_{T,f}$ and $C_{P,f}$, respectively, as the average values of $C_{T,k}$ and $C_{P,k}$ over all the turbines in the farm. Combining Eqs. (5) and (7) into Eq. (8), the farm thrust and power coefficients are thus expressed as

$$C_{T,f} = \frac{\sum_k^{N_t} \overline{u}_{d,k}^2}{N_t \overline{U}_\infty^2} C'_T \text{ and } C_{P,f} = \frac{\sum_k^{N_t} \overline{u}_{d,k}^3}{N_t \overline{U}_\infty^3} C'_T. \quad (9)$$

The analysis is further enriched by considering the farm efficiency $\eta_f$, which can be written in the form of a product of the non-local and the wake efficiencies, denoted $\eta_{nl}$ and $\eta_w$, respectively (Allaerts and Meyers, 2018). We therefore write

$$\eta_f = \eta_{nl}\eta_w, \quad \eta_{nl} = \frac{\overline{P}_1}{\overline{P}_\infty}, \quad \eta_w = \frac{\overline{P}_a}{N_t \overline{P}_1}, \quad (10)$$

where $\overline{P}_a$ is the total farm power measured during the actual simulation, and $N_t$ is the number of turbines in the farm. The notation $\overline{P}_1$ refers to the power per turbine, averaged over the most upstream row in the farm. Finally, $\overline{P}_\infty$ is the power of the turbine operating in isolation. All the quantities in Eq. (10) are time averaged over the 1 h long simulations.

In Fig. 6, we show the thrust coefficients and the power coefficients of an isolated turbine computed following Eq. (8). The results are compared to the expressions

$$C_T = \frac{16C'_T}{\left(C'_T + 4\right)^2} \text{ and } C_P = \frac{64C'_T}{\left(C'_T + 4\right)^3}, \quad (11)$$

obtained from axial momentum theory (AMT) (Allaerts, 2016). Given the time-dependent nature of the thrust and power values collected with a sampling period $T_s = 100$ s,

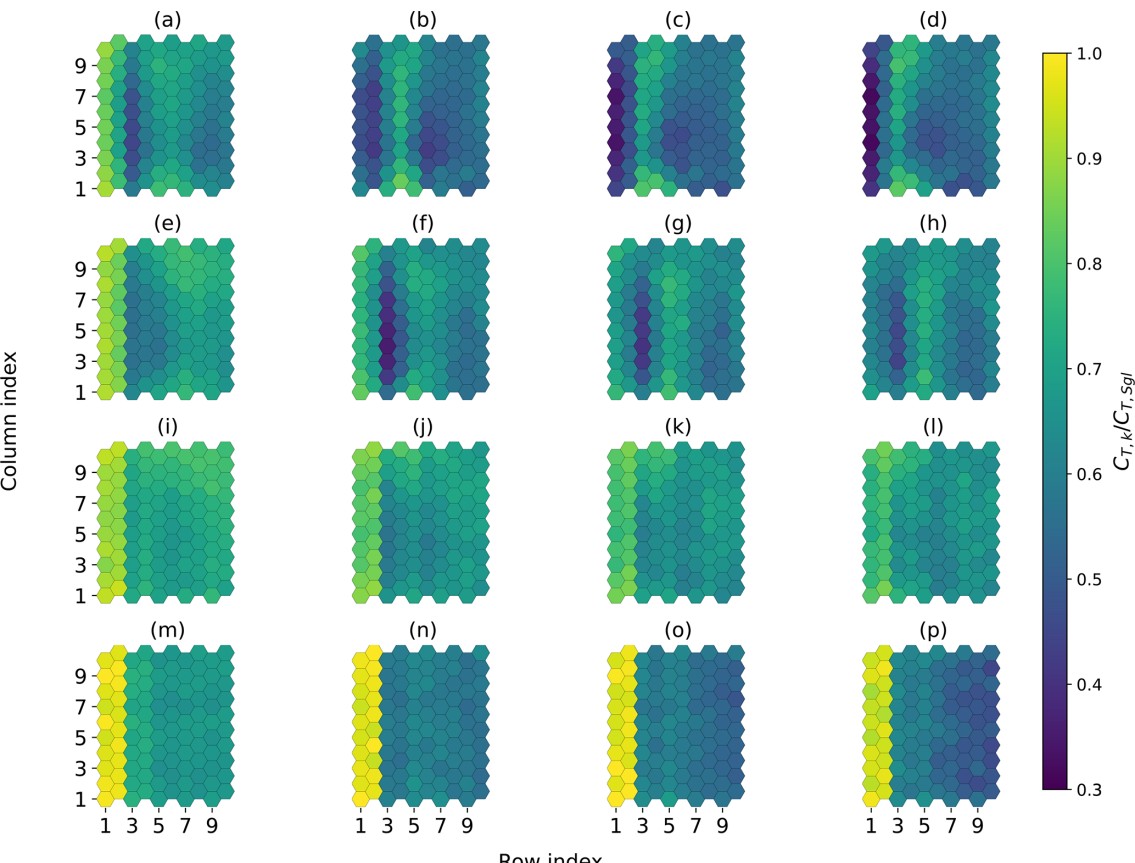

**Figure 5.** Distribution over the farm of the local thrust coefficient (Eq. 8) normalized by the corresponding single-turbine thrust coefficient. The four atmospheric conditions, $H150–\Delta\theta8–\Gamma1$ **(a–d)**, $H300–\Delta\theta5–\Gamma1$ **(e–h)**, $H500–\Delta\theta5–\Gamma4$ **(i–l)** and $H500–\Delta\theta0–\Gamma0$ **(m–p)**, are considered, together with the four operating regimes, $C_T' = 0.50$ **(a, e, i, m)**, $C_T' = 1.25$ **(b, f, j, n)**, $C_T' = 2.0$ **(c, g, k, o)** and $C_T' = 2.75$ **(d, h, l, p)**.

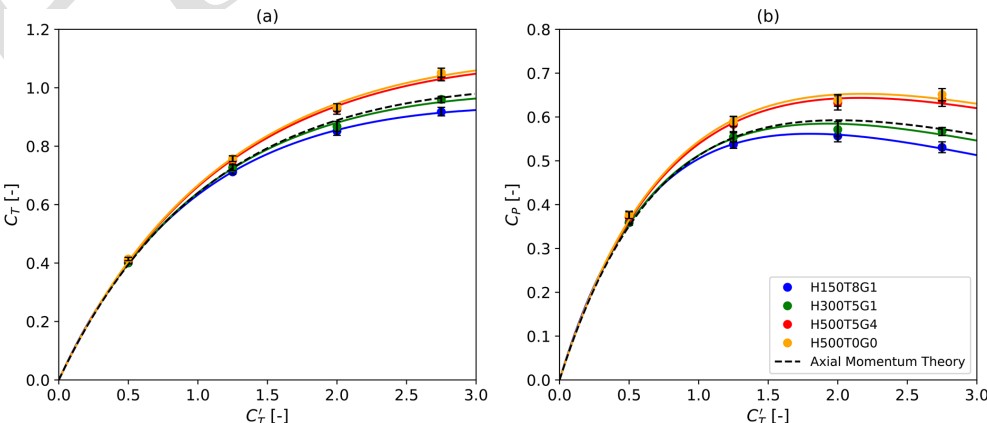

**Figure 6.** Thrust coefficients **(a)** and power coefficients **(b)** as a function of their disk-based counterparts for the standalone wind turbine. The results obtained under the four sets of atmospheric conditions are compared to the predictions of axial momentum theory. The 95 % confidence intervals obtained with the moving-block bootstrap method are shown in black.

the time averages and the 95 % confidence intervals shown in Fig. 6 are computed over the 1 h long actual simulations using a moving-block bootstrap method. We follow the procedure described by Bon and Meyers (2022) with overlapping blocks consisting of $n_\mathrm{b} = 3$ samples over a total of $B = 1000$ bootstrap iterations. We performed a sensitivity study, not discussed here, to motivate the selected values for $n_\mathrm{b}$ and $B$.

As anticipated, the LES results exhibit the same behavior as the theoretical predictions (Eq. 11), in particular at low operating regimes for which close agreement is observed in Fig. 6a and b. Above $C'_\mathrm{T} = 1.25$, the LES values deviate significantly from the AMT for the $H150$–$\Delta\theta8$–$\Gamma1$ and $H500$–$\Delta\theta5$–$\Gamma4$ cases, showing underprediction and overprediction, respectively. Plausible causes of the deviations with respect to classical axial momentum theory include the presence of shear, veer and turbulence in the simulations. It is however unclear whether the overprediction obtained in the $H500$–$\Delta\theta5$–$\Gamma4$ case has a physical explanation. In Shapiro et al. (2019), the authors report that the velocity correction factor leads to slight overestimations at large disk-based thrust coefficients. For instance, we observe a difference of 7 % at $C'_\mathrm{T} = 2.0$ in the present study (Fig. 6b), which aligns with the discrepancy of the order of 5 % retrieved from the results of Shapiro et al. (2019) at the same $C'_\mathrm{T}$ value. Moreover, we emphasize that the expression of the velocity correction factor was obtained for a uniform flow (Shapiro et al., 2019), therefore possibly resulting in larger discrepancies with respect to the AMT when employed in a non-uniform flow. As a matter of fact, the overprediction between the AMT and the results of a uniform-flow simulation performed at $C'_\mathrm{T} = 1.44$ was found to be about half that observed with the $H500$–$\Delta\theta5$–$\Gamma4$ case (not shown). The validity of the velocity correction factor employed with non-uniform profiles could be the topic of future works. The results of LESs relying on a higher-fidelity turbine representation, e.g., an actuator line model, should be used as a reference.

In Fig. 6a and b, we introduce a simple heuristic fit in which the parameters $\alpha_\mathrm{t}$, $\alpha_\mathrm{p}$ and $\beta$ of the laws

$$C_\mathrm{T} = \alpha_\mathrm{t} C'_\mathrm{T} / (\beta C'_\mathrm{T} + 4)^2 \text{ and } C_\mathrm{P} = \alpha_\mathrm{p} C'_\mathrm{T} / (\beta C'_\mathrm{T} + 4)^3 \quad (12)$$

are fitted to the LES data points using the least squares method. As the two relations in Eq. (12) share a common parameter $\beta$, the procedure results in a simple joint-optimization problem between the eight LES data points corresponding to the four tested $C'_\mathrm{T}$ values. The optimized value of each of the three parameters is given in Table 3. In Eq. (12), an increasing number of parameters was introduced, and a convergence analysis on the residual of the least squares method then motivated the choice of $\alpha_\mathrm{t}$, $\alpha_\mathrm{p}$ and $\beta$. Physically, we postulate that the three parameters allow us to account for the impact of shear, veer and turbulence, which are disregarded in classical AMT.

At the wind farm scale, we compute the thrust and power coefficients, $C_\mathrm{T,f}$ and $C_\mathrm{P,f}$, following Eq. (9). Similarly to the

**Table 3.** Values of the three fitting parameters of the single-turbine thrust and power coefficient curves in Fig. 6.

| Case | $\alpha_\mathrm{p}$ | $\alpha_\mathrm{t}$ | $\beta$ |
|------|------|------|------|
| $H150$–$\Delta\theta8$–$\Gamma1$ | 67.3605 | 16.5490 | 1.1104 |
| $H300$–$\Delta\theta5$–$\Gamma1$ | 65.5639 | 16.2539 | 1.0382 |
| $H500$–$\Delta\theta5$–$\Gamma4$ | 64.3789 | 16.0565 | 0.9263 |
| $H500$–$\Delta\theta0$–$\Gamma0$ | 64.7523 | 16.1153 | 0.9184 |

single-wind-turbine case, we employ the moving-block bootstrapping method. However, because the confidence intervals do not exceed $\pm 1$ %, only the time-averaged values of $C_\mathrm{T,f}$ and $C_\mathrm{P,f}$ are represented in Fig. 7. The corresponding curves are fitted using laws of the form

$$C_\mathrm{T,f} = \alpha_\mathrm{t,f} C'_\mathrm{T} / (C'_\mathrm{T} + \delta_\mathrm{t,f})^{\gamma_\mathrm{t,f}} \text{ and}$$

$$C_\mathrm{P,f} = \alpha_\mathrm{p,f} C'_\mathrm{T} / (C'_\mathrm{T} + \delta_\mathrm{p,f})^{\gamma_\mathrm{p,f}}, \quad (13)$$

where 6 degrees of freedom are introduced in total. In Eq. (13), the two sets of three fitting parameters are determined for $C_\mathrm{T,f}$ and $C_\mathrm{P,f}$ through two independent least squares methods. Each of the two fitting procedures therefore sets the values of three parameters using four LES points. The corresponding values are tabulated in Table 4. We initially explored other options, e.g., using only three parameters to approximate $C_\mathrm{T,f}$ and $C_\mathrm{P,f}$ as in the single-turbine case. However, this led to large fitting errors in all the tested cases.

The wind farm thrust and power coefficient curves shown in Fig. 7 can be discussed in parallel to the efficiency curves obtained from Eq. (10) and represented in Fig. 8. First, in Fig. 7b, we notice that the evolution of $C_\mathrm{P,f}$ with $C'_\mathrm{T}$ is much flatter than in the single-wind-turbine situation (Fig. 6b). In Fig. 8c, the farm efficiency is essentially constant above $C'_\mathrm{T} \simeq 1.25$ in the three CNBL cases. This results in a region of nearly constant $C_\mathrm{P,f}$ values, the maximum of which is offset towards $C'_\mathrm{T}$ values lower than in the standalone configuration (Fig. 6b). In the remainder of this analysis, the maximum power coefficient and the corresponding $C'_\mathrm{T}$ and $C_\mathrm{T,f}$ are denoted $C^\star_\mathrm{P,f}$, $C'^\star_T$ and $C^\star_\mathrm{T,f}$, respectively.

Moreover, the inspection of Fig. 8a reveals that the ability of the turbines to generate more power by increasing $C'_\mathrm{T}$ towards its Betz optimal value ($C'_\mathrm{T} = 2$) is considerably harmed by the inevitable blockage effect that accompanies large $C'_\mathrm{T}$ values. This phenomenon appears to be clearly amplified for inflows with a low capping inversion ($H150$–$\Delta\theta8$–$\Gamma1$). On the contrary, the non-local efficiency remains constant with respect to $C'_\mathrm{T}$ in the absence of a capping inversion (case $H500$–$\Delta\theta0$–$\Gamma0$). Under CNBL conditions, we notice in Fig. 8b that the wake efficiency is a growing function of $C'_\mathrm{T}$ that reaches values significantly greater than 1 under specific conditions. This observation is explained by the physical meaning of $\eta_\mathrm{w}$, which should be interpreted as the ratio between the performances of the farm and those of the

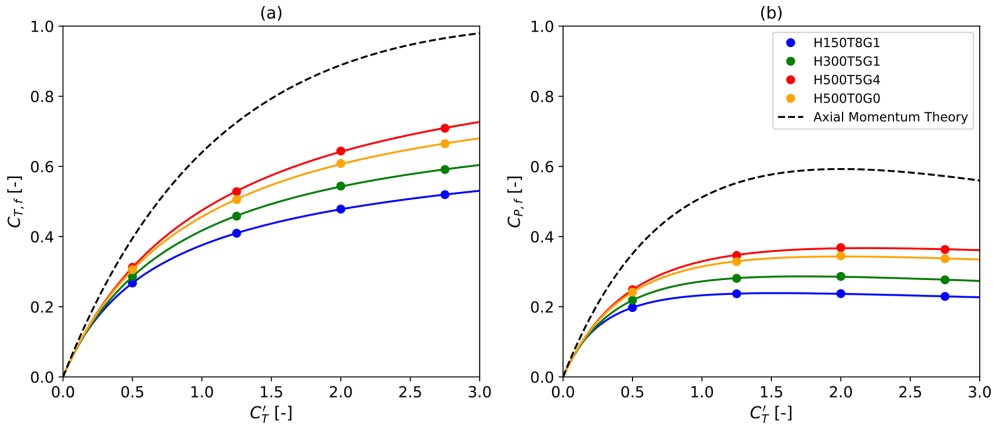

**Figure 7.** Wind farm thrust coefficients **(a)** and power coefficients **(b)** as a function of their disk-based counterparts. The results obtained under the four sets of atmospheric conditions are compared to the predictions of axial momentum theory for a single turbine.

**Table 4.** Values of the six fitting parameters of the wind farm thrust and power coefficient curves in Fig. 7.

| Case | $\alpha_{t,f}$ | $\delta_{t,f}$ | $\gamma_{t,f}$ | $\alpha_{p,f}$ | $\delta_{p,f}$ | $\gamma_{p,f}$ |
|------|------|------|------|------|------|------|
| $H150-\Delta\theta8-\Gamma1$ | 0.5586 | 0.5482 | 0.9080 | 0.4158 | 0.5383 | 1.3484 |
| $H300-\Delta\theta5-\Gamma1$ | 0.7506 | 0.8190 | 0.9820 | 0.6655 | 0.8288 | 1.4803 |
| $H500-\Delta\theta5-\Gamma4$ | 0.9983 | 1.0969 | 1.0040 | 0.9929 | 1.0885 | 1.4982 |
| $H500-\Delta\theta0-\Gamma0$ | 0.9222 | 1.0041 | 1.0113 | 0.9054 | 1.0181 | 1.5057 |

first row. Consequently, the values $\eta_w > 1$ in Fig. 8b correspond to cases where the downstream rows, although waked, extract more power than the first row. This is explained by a large pressure increase before the first row, followed by a favorable pressure gradient that accelerates the flow deeper into the farm, as previously visualized in Fig. 4. Under the atmospheric conditions $H500-\Delta\theta5-\Gamma4$, the favorable pressure gradient leads to values of the farm efficiency that are larger than in the corresponding TNBL case ($H500-\Delta\theta0-\Gamma0$), as indicated in Fig. 8c. Similar observations are reported in Lanzilao and Meyers (2024).

Below $C'_T \simeq 1.25$, the farm poses so little resistance to the flow that only minor blockage effects occur. Simultaneously, we note that this range of operating regimes exhibits high sensitivity to blockage. This is visible in Fig. 8a, where the non-local efficiencies of the $H150-\Delta\theta5-\Gamma1$ case are initially close to those of the $H300-\Delta\theta5-\Gamma1$ and $H500-\Delta\theta5-\Gamma4$ cases but drastically decrease as $C'_T$ increases. This results in a slightly higher farm efficiency at low $C'_T$ values (Fig. 8c), in turn causing the shifting of the curve maximum towards the left in Fig. 7.

Eventually, Fig. 7a shows a strong decrease in the farm thrust coefficient values when compared to those of the isolated turbine (Fig. 6a). However, each wind farm thrust coefficient curve remains much steeper than its power counterpart (Fig. 7), supporting the idea that load can be effectively reduced with only a limited impact on power.

## 4.4  Performance assessment of the collective axial-induction strategy

In this section, the trade-off between thrust and power is explicitly shown in Fig. 9a by plotting the information of Fig. 7a and b in the $C_{P,f}–C_{T,f}$ coordinate system. Figure 9c is obtained by applying the same procedure to the results of the single-turbine simulations (Fig. 6a and b). Then, all the curves in both Fig. 9a and Fig. 9c are normalized by their peak value and represented in Fig. 9b and d, respectively. In Fig. 9d, the normalized curves collapse into the AMT law as this choice of normalization can be shown to be independent of the fitting parameters introduced in Eq. (12). By contrast, we note a clear deviation of the normalized wind farm curves from the predictions of the AMT in Fig. 9b. This observation emphasizes that large-scale effects substantially impact the trade-off between thrust and power and therefore influence the design of the farm operating point. In Fig. 9a, the three curves corresponding to CNBL conditions are affected by both blockage effects and wake interactions. On the contrary, the curve obtained for the $H500-\Delta\theta0-\Gamma0$ case accounts for wake effects only and can thus be considered a blockage-free reference. For each of the curves generated under CNBL conditions in Fig. 9a, we conclude that the deviation observed with respect to the TNBL reference case results from blockage effects.

We now focus on the design of a wind farm operating point that accounts for large-scale effects. To this end, we

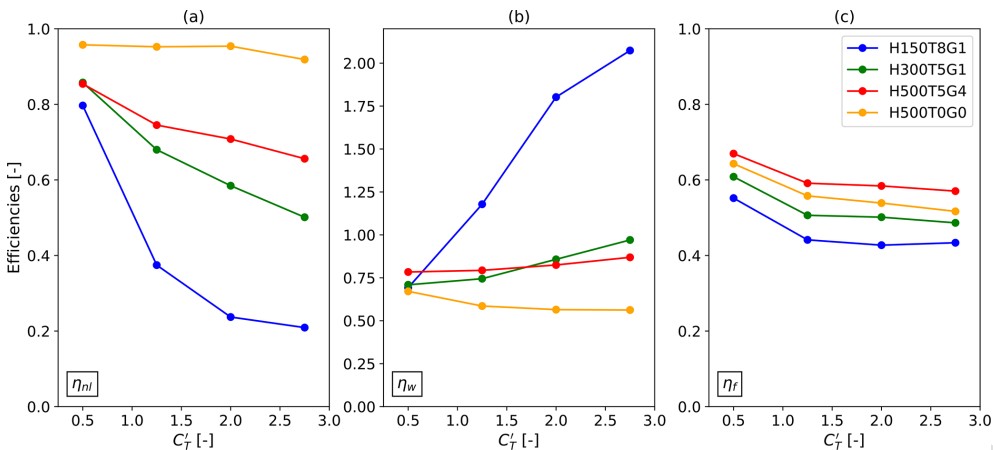

**Figure 8.** Non-local **(a)**, wake **(b)** and global wind farm **(c)** efficiencies computed from Eq. (10) for all the operating points under the four sets of atmospheric conditions. The vertical axis in **(b)** is extended for the sake of readability.

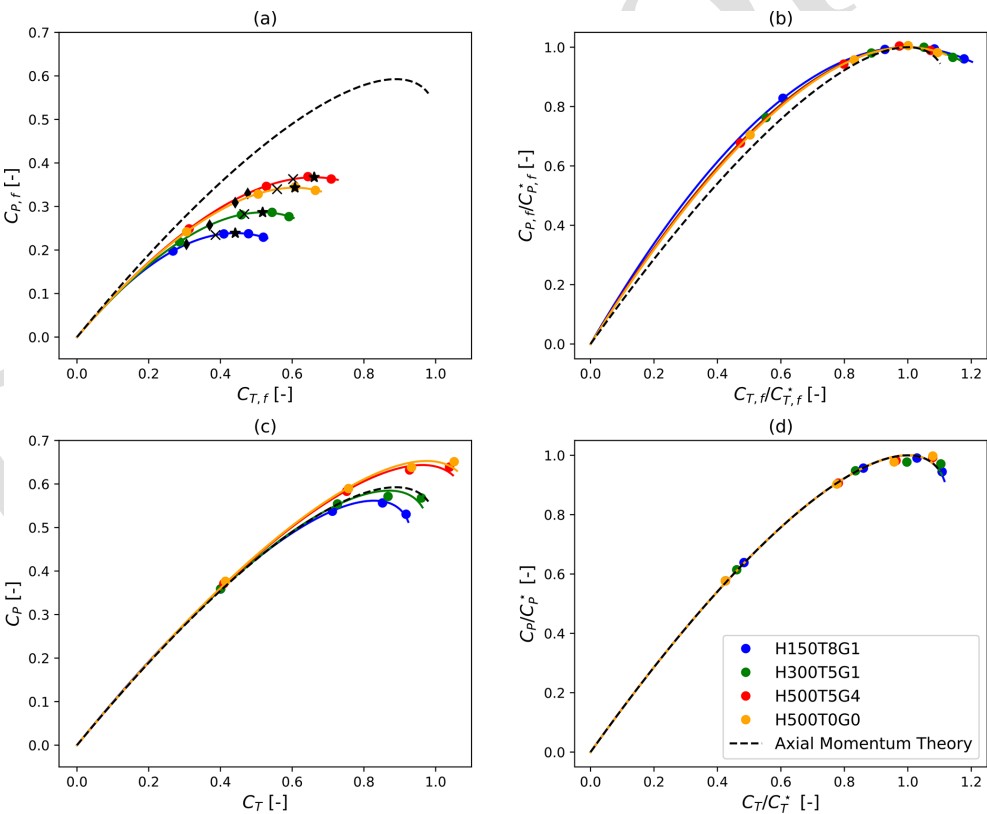

**Figure 9. (a)** Wind farm power coefficient as a function of the wind farm thrust coefficient. For each of the four sets of conditions, the tested operating points $(C_{T,f}^{\star}, C_{P,f}^{\star})$, $(C_{T,f}^{\times}, C_{P,f}^{\times})$ and $(C_{T,f}^{\blacklozenge}, C_{P,f}^{\blacklozenge})$ are indicated with star, cross and diamond symbols, respectively. **(b)** Identical to **(a)** but the maximum farm power coefficient and corresponding thrust coefficient are used to normalize the curve under each condition. **(c)** Single-turbine power coefficient as a function of the thrust coefficient for each condition. **(d)** Identical to **(c)** but the maximum power coefficient and corresponding thrust coefficient are used to normalize the curve under each condition.

explore three potential wind farm set points and evaluate the corresponding thrust and power variation with respect to the standard operating regime $C'_\mathrm{T} = 2$. For the sake of clarity, we denote the coefficients $C_\mathrm{T,f}$ and $C_\mathrm{P,f}$ evaluated at $C'_\mathrm{T} = 2$ by $\hat{C}_\mathrm{T,f}$ and $\hat{C}_\mathrm{P,f}$, respectively. Further, we define the relative thrust and power difference with respect to the standard regime as $\varepsilon_\mathrm{T} = (C_\mathrm{T,f} - \hat{C}_\mathrm{T,f})/\hat{C}_\mathrm{T,f}$ and $\varepsilon_\mathrm{P} = (C_\mathrm{P,f} - \hat{C}_\mathrm{P,f})/\hat{C}_\mathrm{P,f}$.

The first method consists of operating each turbine in the farm at $C'^{\star}_T$ so that the peak of the farm power coefficient curve, i.e., $C^{\star}_\mathrm{P,f}$, is achieved. This operating regime is denoted by a black star in Fig. 9a. An alternative that could be of interest is allowing for a decrease in $C_\mathrm{P,f}$ compared to the standard operating regime $C'_\mathrm{T} = 2$ (Betz limit). With the second approach, we consider for instance a decrease of 1 % in $C_\mathrm{P,f}$. This choice, although somewhat arbitrary, aligns with the reduction in power observed when an isolated IEA 15 MW turbine operates at the design thrust set point, prioritizing load mitigation over maximizing power output (Gaertner et al., 2020). From the three fitted curves shown in Fig. 9a, we retrieve the $C_\mathrm{T,f}$ value at which 99 % of $\hat{C}_\mathrm{P,f}$ is achieved in each case, and we denote it by $C^{\times}_\mathrm{T,f}$. We refer to the corresponding farm power coefficient as $C^{\times}_\mathrm{P,f}$, and we denote the disk-based thrust coefficient by $C'^{\times}_T$. Last, the third method further explores the potential for thrust reduction by allowing for a power decrease of 10 %. Similarly to the second approach, $C^{\blacklozenge}_\mathrm{T,f}$ is the farm coefficient at which 90 % of $\hat{C}_\mathrm{P,f}$ is achieved. The corresponding farm power coefficient and disk-based thrust coefficient are $C^{\blacklozenge}_\mathrm{P,f}$ and $C'^{\blacklozenge}_T$, respectively. The three tested farm operating points and the corresponding gains obtained under each atmospheric condition are listed in Tables 5–7.

In Table 5, we show that operating the farm at $C'^{\star}_T$ to maximize power extraction leads to very slight power increments. This is the case, however, provided that the blockage effect is strong enough, i.e., $H150$–$\Delta\theta8$–$\Gamma1$ and $H300$–$\Delta\theta5$–$\Gamma1$. More interestingly, we observe for those two cases that the power increase, although minor, is associated with a load reduction of the order of 5 %. In the absence of blockage ($H500$–$\Delta\theta0$–$\Gamma0$), the deviation from the standard regime is negligible, indicating that the total power is maximized when each turbine operates at the Betz limit. We anticipate this to no longer be the case in a situation where the $C'_\mathrm{T}$ distribution can be set non-homogeneously across the farm to mitigate the wake effects of the upstream turbines. The results obtained with the second operational strategy are listed in Table 6. From this table, we conclude that substantial load reduction can be achieved at the expense of a minor power loss. In particular, the wind farm thrust coefficients in Table 6 are observed to decrease by up to 19 % under significant blockage. For the same atmospheric conditions, the results of the third approach tabulated in Table 7 indicate a load reduction of 36 %. However, this decrease is limited to 25 % for the $H500$–$\Delta\theta5$–$\Gamma4$ case.

As a conclusion, we show that axial-induction strategies for load reduction are particularly effective for small power reductions relative to the classical operating regime, that is, in the region where the slope of the curves in Fig. 9a is slight. We believe this constitutes an important finding, upon which more sophisticated wind farm operational strategies can be developed. In the future, investigating the sensitivity of the results to the turbine type, the farm layout or the freestream velocity could be of interest. Regarding the ABL flow profile, we anticipate that the diameter-to-hub-height ratio and the ratio of the roughness length to the hub height are meaningful to the problem. We denote them $D^* = D/z_\mathrm{H}$ and $z^*_0 = z_0/z_\mathrm{H}$, respectively. Further, we follow the expression of the similarity parameter $H^* = |f_\mathrm{c}|H/u_*$ (Sood, 2023), where $f_\mathrm{c}$ is the Coriolis frequency, $H$ is the boundary-layer height and $u_*$ is the friction velocity. We note that the hub height velocity $U_\mathrm{H}$ can substitute $u_*$, using a log-law profile and the parameter $z^*_0$ defined above. Throughout the present work, $D^*$ and $z^*_0$ were kept constant, whereas different values of $H^*$ were considered. We refer to, e.g., Csanady (1974), to relate $H^*$ to the potential temperature parameters set in Sect. 2.5. Lastly, the present work provided evidence of the substantial impact of $C'_\mathrm{T}$ on the flow dynamics. More generally, we expect power density to play a crucial part in the design of an effective farm operating strategy. Therefore, we introduce the disk-based friction coefficient factor defined in Calaf et al. (2010) as the fourth non-dimensional number to account for power density. This ratio reads $c'_\mathrm{ft} = \pi C'_\mathrm{T}/(4S_x S_y)$, where $S_x$ and $S_y$ are the turbine spacings (expressed in number of diameters) in the streamwise and spanwise directions, respectively. As a result, similar effects on the total power extraction may be expected for similar values of $c'_\mathrm{ft}$.

## 5   Conclusions

We investigated the potential of collective axial-induction operating strategies in large wind farms to mitigate the effects of blockage. For this purpose, a series of large-eddy simulations of a large farm of 100 IEA 15 MW turbines placed in a staggered configuration was performed. Each turbine was represented by an actuator disk with an adjustable disk-based thrust coefficient. Overall, the study covered three different conventionally neutral boundary-layer conditions, for which little to strong blockage effects were expected. Additionally, a fourth set of atmospheric conditions was considered, representing a truly neutral boundary layer. Alongside varying the flow conditions, the disk-based thrust coefficient of each turbine in the farm was successively set to four values uniformly over the farm. Consequently, a total of 16 simulations were carried out.

First, a precursor simulation was run for each of the three CNBL conditions, after which a spin-up simulation was performed for every operating regime. A convergence analysis on the farm power motivated the use of 90 min long spin-up

**Table 5.** Operating parameters selected in the first approach and corresponding gains with respect to the classical operating point for the four atmospheric conditions. The disk-based thrust coefficient is set to $C'^{\star}_T$ to maximize power extraction.

| | $H150-\Delta\theta8-\Gamma1$ | $H300-\Delta\theta5-\Gamma1$ | $H500-\Delta\theta5-\Gamma4$ | $H500-\Delta\theta0-\Gamma0$ |
|---|---|---|---|---|
| $C'^{\star}_T$ [–] | 1.55 | 1.73 | 2.18 | 2.01 |
| $C^{\star}_{P,f}$ [–] | 0.24 | 0.29 | 0.37 | 0.34 |
| $C^{\star}_{T,f}$ [–] | 0.44 | 0.52 | 0.66 | 0.61 |
| $\varepsilon_P$ [%] | 0.83 | 0.35 | 0.13 | 0.001 |
| $\varepsilon_T$ [%] | −7.64 | −4.59 | 3.07 | 0.22 |

**Table 6.** Operating parameters selected in the second approach and corresponding gains with respect to the classical operating point for the four atmospheric conditions. The disk-based thrust coefficient is set to $C'^{\times}_T$ so that 99 % of $\hat{C}_{P,f}$ is achieved.

| | $H150-\Delta\theta8-\Gamma1$ | $H300-\Delta\theta5-\Gamma1$ | $H500-\Delta\theta5-\Gamma4$ | $H500-\Delta\theta0-\Gamma0$ |
|---|---|---|---|---|
| $C'^{\times}_T$ [–] | 1.07 | 1.30 | 1.69 | 1.58 |
| $C^{\times}_{P,f}$ [–] | 0.23 | 0.28 | 0.36 | 0.34 |
| $C^{\times}_{T,f}$ [–] | 0.39 | 0.47 | 0.6 | 0.56 |
| $\varepsilon_P$ [%] | −1.0 | −1.0 | −1.0 | −1.0 |
| $\varepsilon_T$ [%] | −19.16 | −14.01 | −6.1 | −7.98 |

phases. In each case, thrust and power measurements were subsequently collected over a 1 h long simulation.

The streamwise velocity fields provided evidence of the significant mesoscale effects induced by the presence of the farm and shed light on the conditions that foster these effects. For low-capping-inversion cases, a low-velocity region was observed to develop upstream of the farm in the form of a bow wave. However, we showed that this blockage effect was significantly attenuated for low values of the disk-based thrust coefficients. Next, the analysis of the thrust distribution throughout the farm indicated strong heterogeneities caused by the simultaneous effects of wakes and blockage.

The results were then discussed in terms of the wind farm thrust and power coefficients, together with the wind farm, wake and non-local efficiencies. For all the tested CNBL conditions, the non-local efficiency decreased with increasing $C'_T$, with a significant drop observed for the $H150-\Delta\theta8-\Gamma1$ case in particular. For the same conditions, wake efficiencies greater than 1 further indicated the presence of a favorable pressure gradient throughout the farm. For values of the disk-based thrust coefficient larger than 1.25, the farm efficiency was found to be essentially constant with $C'_T$ but strongly dependent on the atmospheric conditions. As a result, we observed a flattening of the farm power coefficient curve with respect to its single-turbine counterpart. Finally, we proposed three approaches to address thrust and power trade-offs. We found that operating the turbines below the optimal Betz point could simultaneously maximize power extraction and reduce the loading by more than 7 % under strong blockage. We further concluded that enabling a 1 % power reduction could result in a load decrease of 6 % to 19 %, depending on the conditions. The same factor was seen to reach between 25 % and 36 % at the expense of a power decrease of 10 %, however.

In the future, we plan on expanding the study to other values of the capping inversion parameters. More generally, a similar analysis performed for stable and unstable boundary-layer profiles could be of interest. Finally, we intend to investigate the benefits of more advanced operational strategies, for instance by considering non-uniform $C'_T$ distributions over the farm.

## Appendix A: Detailed formulation of the governing equations

The set of equations described in Sect. 2.1 for the three-dimensional filtered velocity field ($u_i$) and the filtered potential temperature ($\theta$) reads as follows:

$$\frac{\partial u_i}{\partial x_i} = 0, \tag{A1}$$

$$\frac{\partial u_i}{\partial t} + \frac{\partial}{\partial x_j}(u_j u_i) = f_c \epsilon_{ij3} u_j + \delta_{i3} g \frac{\theta - \theta_0}{\theta_0} - \frac{\partial \tau^{sgs}_{ij}}{\partial x_j}$$
$$- \frac{1}{\rho_0} \frac{\partial p^*}{\partial x_i} - \frac{1}{\rho_0} \frac{\partial p_\infty}{\partial x_i} + f^{tot}_i, \tag{A2}$$

$$\frac{\partial \theta}{\partial t} + \frac{\partial}{\partial x_j}(u_j \theta) = -\frac{\partial q^{sgs}_j}{\partial x_j}, \tag{A3}$$

where Eqs. (A1)–(A3) are the continuity, momentum and potential temperature transport equations, respectively. Note that the streamwise, spanwise and vertical directions are indicated by the indices $i = 1$, 2 and 3, respectively. In Eq. (A2), the first term on the right-hand side accounts for the Cori-

**Table 7.** Operating parameters selected in the third approach and corresponding gains with respect to the classical operating point for the four atmospheric conditions. The disk-based thrust coefficient is set to $C'^{\blacklozenge}_T$ so that 90 % of $\hat{C}_{P,f}$ is achieved.

| | $H150–\Delta\theta8–\Gamma1$ | $H300–\Delta\theta5–\Gamma1$ | $H500–\Delta\theta5–\Gamma4$ | $H500–\Delta\theta0–\Gamma0$ |
|---|---|---|---|---|
| $C'^{\blacklozenge}_T$ [–] | 0.64 | 0.78 | 1.01 | 0.93 |
| $C^{\blacklozenge}_{P,f}$ [–] | 0.21 | 0.26 | 0.33 | 0.31 |
| $C^{\blacklozenge}_{T,f}$ [–] | 0.31 | 0.37 | 0.48 | 0.44 |
| $\varepsilon_P$ [%] | −10.0 | −10.0 | −10.0 | −10 |
| $\varepsilon_T$ [%] | −36.11 | −31.84 | −25.81 | −27.25 |

olis force generated by the rotation of the Earth at angular velocity $\Omega_E$ and latitude $\phi$, where $f_c = 2\Omega_E \sin\phi$ is the Coriolis frequency and $\epsilon_{ij3}$ the Levi–Citiva symbol. Further, the buoyancy effect on the vertical momentum is represented by the second component in the right-hand side term of Eq. (A2), where $\theta_0$ denotes the reference potential temperature, and $\delta_{i3}$ is the Kronecker delta. The effect of the subgrid-scale dynamics and heat transfer on the resolved flow is accounted for through the stress tensor $\tau^{sgs}_{ij}$ (Eq. A2) and the heat flux $q^{sgs}_j$ (Eq. A3), respectively. In Eq. (A2), the background pressure and the filtered fluctuations around it are denoted $p_\infty$ and $p^*$. Eventually, the body force term $f^{tot}_i$ is composed of the wind farm forcing on the flow (Eq. 6), the fringe region forcing (Eq. B3) and the Rayleigh damping (Eq. B1).

## Appendix B: Mathematical expressions of the Rayleigh damping, the fringe forcing functions and the vertical momentum damping factor

As introduced in Sect. 2.3, a Rayleigh damping layer is used as the non-reflective upper-boundary condition in the main domain. Along the three directions ($i = 1, 2, 3$), the corresponding forcing term per unit mass reads

$$f^{ra}_i(\boldsymbol{x}) = -\nu(z)(u_i(\boldsymbol{x}) - U_{g,i}), \quad (B1)$$

where $U_{g,i}$ is the component of the geostrophic wind $G$ along the considered direction. The buffer intensity increases with height at a rate controlled by the Rayleigh function $\nu(z)$. Following Lanzilao and Meyers (2023), we write for $z > (L_z - L^{ra}_z)$

$$\nu(z) = \breve{\nu}\left(1 - \cos\left(\frac{\pi}{s^{ra}}\frac{z - (L_z - L^{ra}_z)}{L^{ra}_z}\right)\right), \quad (B2)$$

with $\breve{\nu} = \nu^{ra}N$ being the amplitude parameter and $N$ the Brunt–Väisälä frequency. The values of $L^{ra}_z$, $\nu^{ra}$ and $s^{ra}$ (see Table 1) are determined in light of the thorough analysis provided by Lanzilao and Meyers (2023) to minimize reflectivity.

Moreover, in Lanzilao and Meyers (2023), the forcing term related to the fringe region is expressed as

$$f^{fr}_i(\boldsymbol{x}) = -h(x)(u_i(\boldsymbol{x}) - u_{prec,i}(\boldsymbol{x})), \quad (B3)$$

where $u_{prec,i}$ denotes the velocity field retrieved from the concurrent precursor simulation. To ensure that the forcing is gradually applied over the fringe region, we employ the smooth function

$$h(x) = -h_{max}\left(F\left(\frac{x - x^h_s}{\delta^h_s}\right) - F\left(\frac{x - x^h_e}{\delta^h_e} + 1\right)\right), \quad (B4)$$

where

$$F(x) = \begin{cases} 0, & \text{if } x \le 0 \\ \frac{1}{1+\exp\left(\frac{1}{x-1} + \frac{1}{x}\right)} & \text{if } 0 < x < 1 \\ 1, & \text{if } x \ge 1. \end{cases} \quad (B5)$$

The values of the parameters $x^h_s$, $x^h_e$, $\delta^h_s$ and $\delta^h_e$ are given in Table 2. Finally, Lanzilao and Meyers (2023) propose locally damping the vertical momentum term in the fringe region so as to prevent the propagation of gravity waves triggered by the fringe forcing. The damping factor multiplies the vertical momentum convective term in Eq. (A2) and is expressed as

$$d(x, z) = 1 - \left(F\left(\frac{x - x^d_s}{\delta^d_s}\right) - F\left(\frac{x - x^d_e}{\delta^d_e} + 1\right)\right) \\ \times \mathcal{H}(z - H), \quad (B6)$$

where the Heaviside function $\mathcal{H}$ ensures zero damping inside the ABL, i.e., up to $H$. The selected values of the parameters $x^d_s$, $\delta^d_s$, $x^d_e$ and $\delta^d_e$ in Eq. (B6) are tabulated in Table 2.

**Code availability.** TS5

**Data availability.** TS6

**Author contributions.** TD and JM jointly defined the methodology and the simulation set-ups. The simulations and post-processing steps were carried out by TD. TD and JM jointly wrote the paper.

**Competing interests.** At least one of the (co-)authors is a member of the editorial board of *Wind Energy Science*. The peer-review

process was guided by an independent editor, and the authors also have no other competing interests to declare.

**Disclaimer.** Publisher's note: Copernicus Publications remains neutral with regard to jurisdictional claims made in the text, published maps, institutional affiliations, or any other geographical representation in this paper. While Copernicus Publications makes every effort to include appropriate place names, the final responsibility lies with the authors.

**Acknowledgements.** The authors gratefully acknowledge support from the Belgian Federal Public Planning Service Science Policy (BELSPO). The computational resources and service in this work were provided by the Flemish Supercomputer Center (VSC), funded by the Research Foundation Flanders (FWO) and the Flemish Government Department of EWI. The authors thank Luca Lanzilao for helpful discussions.

**Financial support.** This work was done under project ETREND, funded by the Belgian Federal Public Planning Service Science Policy (BELSPO) under the Brain-be 2.0 program (contract number B2/223/P1/E-TREND). TS7

**Review statement.** This paper was edited by Cristina Archer and reviewed by two anonymous referees.

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

## Remarks from the typesetter