# Peer review of "A large-eddy simulation analysis of collective wind farm axial-induction set points in the presence of blockage"

_Wind Energy Science, 2024_

## Author Comment (AC2)

**Response to the reviewers**

We thank the reviewers for their critical assessment of our work. In the following we address their concerns point by point that in our opinion has substantially improved the manuscript. The reply below gives a motivation of the changes, as well as the additions to the manuscript (written *in italic*, between the last and the first unchanged sentence in the text). The figure numbers in the reply refer to the figures in the improved manuscript.
* * *
**Reviewer 1**

This paper is of sure interest for the field of wind energy and very well written. The results appear robust and original. However, I have a few concerns that should be addressed by the authors:

**Reviewer Point P 1.1** — p.5: Numerical set-up: Please define the set of equations used in your LES (I guess NS equations with Coriolis and temperature stratification effect with Boussinesq approximation for the potential temperature as in Allaerts & Meyers 2015). Also some of the used techniques (i.e., the fringe region, the tilting technique, etc) should be discussed in more detail for making the paper more self-contained.

**Reply**: We thank you for this suggestion. The set of equations is now given and explained in "Appendix A: Detailed formulation of the governing equations". With regard to the fringe region technique, further details are now provided in "Appendix B: Mathematical expressions of the Rayleigh-damping, the fringe forcing functions and the vertical momentum-damping factor". Clarifications about the tiling technique are also added in Sec. 2.4.

- *The set of equations described in Sec. 2.1 for the three-dimensional filtered velocity field ($u_i$) and the filtered potential temperature ($\theta$) reads as follows:*

$$\frac{\partial u_i}{\partial x_i} = 0 \tag{1}$$

$$\frac{\partial u_i}{\partial t} + \frac{\partial}{\partial x_j}(u_j u_i) = f_c \epsilon_{ij3} u_j + \delta_{i3} g \frac{\theta - \theta_0}{\theta_0} - \frac{\partial \tau_{ij}^{sgs}}{\partial x_j} - \frac{1}{\rho_0}\frac{\partial p^*}{\partial x_i} - \frac{1}{\rho_0}\frac{\partial p_\infty}{\partial x_i} + f_i^{tot} \tag{2}$$

$$\frac{\partial \theta}{\partial t} + \frac{\partial}{\partial x_j}(u_j \theta) = -\frac{\partial q_j^{sgs}}{\partial x_j}, \tag{3}$$

*where Eq. 1, 2 and 3 are the continuity, momentum and potential-temperature transport equations, respectively. Note that the streamwise, spanwise and vertical directions are indicated by the indices i = 1,2 and 3, respectively, while the tilde refers to the filtered quantities. In Eq. 2, the first term on the right-hand side accounts for the Coriolis force generated by the rotation of the earth at angular velocity $\Omega_E$ and latitude $\phi$, where $f_c = 2\Omega_E \sin\phi$ is the Coriolis frequency and $\epsilon_{ij3}$ the Levi-Citiva symbol. Further, the buoyancy effect on the vertical momentum is represented by the second component in the RHS term of Eq. 2, where $\theta_0$ denotes the reference potential temperature and $\delta_{i3}$ is the Kronecker delta. The effect of the subgrid-scale dynamics and heat transfer on the resolved flow is accounted for through the stress tensor $\tau_{ij}^{sgs}$ (Eq. 2) and the heat flux $q_j^{sgs}$ (Eq. 3), respectively. In Eq. 2, the background pressure and the filtered fluctuations around it are denoted $p_\infty$ and $p^*$. Eventually, the body force term $f_i^{tot}$ is composed of the wind-farm forcing on the flow, the fringe-region forcing and the Rayleigh damping.*

- *As introduced in Sec. 2.3, a Rayleigh-damping layer is used as the non-reflective upper boundary condition in the main domain. Along the three directions ($i = 1, 2, 3$) the corresponding forcing term per unit mass reads:*

$$f_i^{ra}(\mathbf{x}) = -\nu(z)\left(u_i(\mathbf{x}) - U_{g,i}\right),\qquad(4)$$

  *where $U_{g,i}$ is the component of the geostrophic wind $G$ along the considered direction. The buffer intensity increases with height at a rate controlled by the Rayleigh function $\nu(z)$. Following Lanzilao & Meyers (2023), we write for $z > (L_z - L_z^{ra})$:*

$$\nu(z) = \breve{\nu}\left(1 - \cos\left(\frac{\pi}{s^{ra}}\frac{z - (L_z - L_z^{ra})}{L_z^{ra}}\right)\right),\qquad(5)$$

  *with $\breve{\nu} = \nu^{ra}N$ the amplitude parameter and $N$ the Brunt-Väisälä frequency. The values of $L_z^{ra}$, $\nu^{ra}$ and $s^{ra}$ (see Table. 1) are determined in light of the thorough analysis provided by Lanzilao & Meyers (2023) to minimize reflectivity. Moreover, in Lanzilao & Meyers (2023), the authors express the forcing term related to the fringe-region as:*

$$f_i^{fr}(\mathbf{x}) = -h(x)\left(u_i(\mathbf{x}) - u_{prec,i}(\mathbf{x})\right),\qquad(6)$$

  *where $u_{prec,i}$ denotes the velocity field retrieved from the concurrent precursor simulation. To ensure that the forcing is gradually applied over the fringe-region, we employ the smooth function*

$$h(x) = -h_{max}\left(F\left(\frac{x - x_s^h}{\delta_s^h}\right) - F\left(\frac{x - x_e^h}{\delta_e^h} + 1\right)\right),\qquad(7)$$

  *where*

$$F(x) = \begin{cases} 0, & if\ x \le 0 \\ \frac{1}{1+\exp\left(\frac{1}{x-1}+\frac{1}{x}\right)} & if\ 0 < x < 1 \\ 1, & if\ x \ge 1. \end{cases}\qquad(8)$$

  *The values of the parameters $x_s^h$, $x_e^h$, $\delta_s^h$, $\delta_e^h$ are given in Table. 2. Finally, Lanzilao & Meyers (2023) propose to locally damp the vertical momentum term in the fringe region so as to prevent the propagation of gravity waves triggered by the fringe forcing. The damping factor multiplies the vertical momentum convective term in Eq. 2 and is expressed as:*

$$d(x,z) = 1 - \left(F\left(\frac{x - x_s^d}{\delta_s^d}\right) - F\left(\frac{x - x_e^d}{\delta_e^d} + 1\right)\right)\mathcal{H}(z - H),\qquad(9)$$

  *where the Heaviside function $\mathcal{H}$ ensures zero damping inside the ABL, i.e. up to $H$. The selected values of the parameters $x_s^d$, $\delta_s^d$, $x_e^d$ and $\delta_e^d$ in Eq. 9 are tabulated in Table. 2.*

- ...Eventually, the distance upstream of the farm is taken equal to $L_{\text{ind}} = 18\,\text{km}$ to allow for a full representation of the induction zone (Lanzilao & Meyers, 2024).
  *Because periodicity is imposed over the four lateral sides of the precursor domain, the tiling technique of Sanchez Gomez et al. (2023) is employed to extend the ten-kilometer-long and ten-kilometer-wide precursor field to the horizontal dimensions of the main domain. The resulting field is used as the initial state in the wind-farm simulations. The same tiling operation, limited to the spanwise direction however, is carried out to generate the concurrent*

*precursor with horizontal dimensions $L_x^{cp} \times L_y^{cp} = 10 \times 40\,km^2$. Additionally, we artificially extend the height of the precursor field by imposing the geostrophic flow field from 3 km to 25 km for all the considered atmospheric conditions, i.e. in the region where the flow field can reasonably be assumed laminar. The characteristics of the precursor field are further discussed in Sec. 3.1.*

Furthermore, the grid resolution is identical to that selected by Lanzilao & Meyers (2024),...

**Reviewer Point P 1.2** — p5: the Smagorinsky coefficient is rather high for wind turbine simulations. Please justify the choice of this value with respect to the literature.

**Reply**:  The value of the Smagorinsky coefficient is identical in previous works carried out with SP-Wind: Lanzilao & Meyers (2023, 2024), Munters & Meyers (2018), Allaerts & Meyers (2017) and Goit & Meyers (2015). Those references are now added in the text. Moreover, we emphasize that we employ the damping technique of Mason and Thomson (1992) near the wall.

... .The corresponding Smagorinsky coefficient is set to $C_s = 0.14$, similar to previous works carried out with SP-Wind *(Goit and Meyers, 2015; Allaerts and Meyers, 2017; Munters and Meyers, 2018 and Lanzilao and Meyers (2023, 2024)). Moreover, we use the damping approach of Mason and Thomson (1992) near the wall, which is a well established technique for neutral ABL's (see also Meyers, 2011).* In order to solve the set of equations, we use the in-house SP-WInd solver...

**Reviewer Point P 1.3** — p5: concerning the grid, in the spanwise direction there is one grid point every 22 meters, while much finer cells are used for the vertical direction. The fact that cells are so elongated in the spanwise direction might constitute a numerical issue. Please validate and/or justify with respect to previous works.

**Reply**:

We thank you for bringing up this relevant point. As the grid resolution was chosen identical to that of Lanzilao & Meyers (2024), we did not carry out any grid sensitivity analysis. However, based on our experience with SP-Wind, aspect ratios of the order 3-to-4 have been observed to perform well, sometimes better than an aspect ratio of 2. This is explained by the differences between the pseudo-spectral Fourier scheme used along $y$ and the energy-preserving fourth-order finite difference scheme employed vertically.

... .Note that the same vertical discretization, trimmed to $L_z^p = 3\,km$, however, is adopted for the initial precursor simulation. *In the region where the vertical grid spacing is the finest, the spanwise-vertical aspect ratio is equal to $\Delta y/\Delta z \simeq 4.3$. Although no detailed study on the aspect ratio impact was performed with SP-Wind, values of the order 3-to-4 were historically retained (Allaerts & Meyers, 2017; Lanzilao & Meyers (2023,2024)) in order to account for the differences between the discretization schemes used in the spanwise and vertical directions (see Sec. 2.3).* In Sec. 4.2, the power production of an isolated turbine is compared to that of the wind farm for reference.

**Reviewer Point P 1.4** — p.5: "the authors consider a wind farm about 4km longer.... we use the same main domain length". If in Lanzilao & Meyers (2024) the domain was 4km longer, how can you use the same domain length? Please explain.

**Reply**:

In the work of Lanzilao & Meyers (2024), the considered wind-farm is 4km longer than the wind-farm we study in the present paper. However, we keep the same domain length, *i.e.* 50km, as in Lanzilao & Meyers (2024). The dimensions of the domain employed in Lanzilao & Meyers (2024) are now clearly specified to avoid ambiguities (Sec. 2.4).

...We note that in the latter study, the authors consider a wind farm about $4\,\mathrm{km}$ longer but $2\,\mathrm{km}$ narrower than the one investigated in the present work (see Sec. 2.5). *Therefore, we use the same main domain length and height as in Lanzilao & Meyers (2024), i.e. $L_x \times L_z = 50 \times 25\,km^2$, but we increase the domain width by 10km so that the main domain has dimensions $L_x \times L_y \times L_z = 50 \times 40 \times 25\,km^3$. While the domain height may initially appear overly large, it is required to allow for the non-reflecting radiation of gravity waves and to accommodate the Rayleigh damping layer described in Sec. 2.3.* The farm is symmetrically positioned along the spanwise direction, resulting in a distance of $L_{\mathrm{side}} = 14.3\,\mathrm{km}$ between the edges of the farm and the lateral sides of the domain...

**Reviewer Point P 1.5** — -p.5 "we artificially extend the height of the precursor field by imposing the geostrophic flow field from 3km to 25 km..." This technique is questionable, since turbulent fluctuations are interrupted abruptly and may induce non physical effects. In fact, the flow field will be subject to abrupt changes, which can affect the results. The authors should clearly show and discuss what happens in the region where this strong discontinuity is imposed, by plotting rms quantities or Reynolds stresses. If a strong discontinuity on these quantities is indeed present, they should consider performing a computation adding a smoothing of the turbulent fluctuations instead of a discontinuity and show that this has virtually no effect on the results.

**Reply**: For all the considered conditions, the ABL grows during the precursor simulation but remains much shallower than 1 km (as observed in Fig. 2 (a,b)). Above the boundary layer, the flow is laminar. Any inertial fluctuations or gravity waves that form in the free-atmosphere during the precursor simulation are damped by mean of a Rayleigh-damping layer applied above 1 km. This is now explained in the text (Sec. 3.1) as follows:

- ...The initial potential temperature profiles are generated using the Rampanelli & Zardi (2004) model together with the sets of parameters, H, $\Delta\theta$ and $\Gamma$, detailed in Sec. 2.5. *We emphasize that a Rayleigh-damping layer is applied during the precursor phase to damp the inertial fluctuations and the gravity waves above 1 km in the atmosphere.* For each atmospheric condition, the precursor simulation is performed over 20 hours...

- ...the super-geostrophic jet that forms at the top of the ABL increases with decreasing inversion-layer heights. *Above the jet, the shear stress profile reduces to zero (Fig. 2 (b)), the flow becomes laminar, and the velocity profile corresponds to the geostrophic wind.* Figure 2 (d) shows that for the cases H150-$\Delta\theta$8-$\Gamma$1, H300-$\Delta\theta$5-$\Gamma$1, H500-$\Delta\theta$5-$\Gamma$4...

**Reviewer Point P 1.6** — Figure 2: is the vertical axis indeed in [km]? It is weird to see that the capping inversion is so low while the the domain extends 25 km in the vertical direction. Moreover, please confirm that the laminar geostrophic flow field is added starting from 3km for all the three cases independently of the location of the capping inversion.

**Reply**: We confirm that the vertical axis in Figure 2 is expressed in [km]. The domain height is set so that gravity waves can develop aloft, while ensuring minimum reflection at the top (Rayleigh-damping layer). Moreover, we verify that, in each case, the boundary-layer height has the correct order of magnitude. In Csanady (1974), the height of the inversion-layer $H$ in equilibrium conditions verifies $g'H/(Au_\star^2) \simeq 1$, where $u_\star$, $g'$ and $A$ are the friction velocity, the reduced gravity and an empirical constant equal to 500, respectively. For the case $H150$-$\Delta\theta8$-$\Gamma1$ considered in the present work, we therefore compute $H \simeq 140\,\mathrm{m}$. This result is of the same order of magnitude as the inversion-layer height shown in Figure 2 in the case $H150$-$\Delta\theta8$-$\Gamma1$, *i.e.* $H \simeq 195\,\mathrm{m}$. We re-write the beginning of the paragraph starting at line 215 to discuss this matter. Finally, we confirm that the laminar geostrophic flow field is added starting from 3km for all the three cases and we specify it in the manuscript.

- ...From Fig. 2 (a), it can be seen that the presence of the capping inversion limits the boundary layer growth, *so that the equilibrium inversion-layer height is attained when buoyancy forces balance the surface shear stress (Csanady 1974). As observed by Lanzilao & Meyers (2024), the amplitude of the super-geostrophic jet that forms at the top of the ABL increases with decreasing inversion-layer heights.* Above the jet, the shear stress profile reduces to zero...

- ...the origin of the capping inversion moves to an altitude of 195 m, 325 m and 510 m, respectively, over the twenty-hour-long spin-up. *We note that these values align with the predictions computed from Csanady (1974) (not detailed here).* In the ABL, the Ekman spiral forms so that the wind-direction angle...

**Reviewer Point P 1.7** — - Figure 2 c: in the region where the flow angle becomes constant, there are oscillations of the flow angle. Please justify its origin and its effect on the flow.

**Reply**: We thank the reviewer for this comment and agree that it is worth further discussion. Although the source of those oscillations has not been formally shown in SP-Wind, we postulate that they result from the use of the stratification-independent SGS model in a region where some of the non-resolved eddies are stratification-dependent. This is now discussed using literature as follows:

...Note that the wind-direction controller designed by Allaerts & Meyers (2015) is employed during the precursor phase to rotate the geostrophic wind so as to ensure no spanwise velocity components at hub height, i.e. $\Phi(z_{hub}) = 0°$. *Finally, small oscillations of the velocity magnitude and the wind direction appear in the inversion layer (Fig. 2 (a,b)), as a result of the strong stratification that characterizes this region. This matter is addressed in Sullivan et al. (2016), where the authors show that eddies with a characteristic scale larger than the Doughtery–Ozmidov length are stratification-dependent. However, this length decreases as stratification increases, possibly leading to values of the Doughtery–Ozmidov length smaller than the grid spacing. Some of the Sub-grid*

*scale eddies generated in the inversion-layer can be stratification dependent and can therefore not be accurately captured by the SGS model, causing the oscillations observed in Fig. 2 (a,b). Similar oscillations can be seen in e.g. Maas and Raasch (2022) and Pedersen et al. (2014).*

**Reviewer Point P 1.8** — p.6: please discuss the choice of the $C_T$ values, are those typical for a 15MW IEA wind turbine in which operating conditions?

**Reply**: The $C_T'$ values have been selected to sample the set of possible values (i.e. 0 to 4) with an arbitrary spacing of 0.75 and include the Betz limit ($C_T' = 2$). This is now explained in the paper as:

...In order to explore the potential for power optimization and load reduction using axial induction control, different disc-based thrust coefficients (Eq. 5) are tested. *From axial momentum theory, the value $C_T = 8/9$, or equivalently $C_T' = 2$ (Allaerts 2016), maximizes power extraction. In practice, the designed thrust-set-point is slightly lower to reduce the associated loads at rated wind speed (Gaertner et al. 2020). In the scope of this work, we therefore select the theoretical optimal disc-based thrust coefficient, along with three other values of $C_T'$ evenly spaced at intervals of 0.75: $C_T' = \{0.50; 1.25; 2.0; 2.75\}$. Following classical momentum theory (Allaerts 2016), the corresponding values of the thrust coefficient are $C_T = \{0.40; 0.73; 0.89; 0.97\}$.* We emphasize that in all the simulations, the considered $C_T'$ value is constant throughout the wind plant,...

**Reviewer Point P 1.9** — p7: "the wind-direction controller designed by Allaerts and Meyers (2015) is employed during the precursor phase" Is the controller active only during the precursor simulation or also in the rotor simulations?

**Reply**: The wind-angle controller of Allaerts & Meyers in employed in the precursor and spin-up simulations but turned off in the actual simulations. This is re-phrased in the text as follows.

...The power and flow quantities are measured during this last phase, referred to as the actual simulation. *Similar to the precursor phase (Sec. 3.1), the wind-direction controller of Allaerts & Meyers (2015) is employed during the wind farm spin-up phase. This controller is disabled in the actual simulation, however.* Eventually, the same procedure is applied to the corresponding single wind turbine cases over the small domain, as discussed in Sec. 2.4

**Reviewer Point P 1.10** — p16: "clearly amplified for inflows with a low capping inversion" Please discuss, with reference to the literature, the behaviour in the absence of a capping inversion.

**Reply**: We agree with the reviewer that a comparison with a truly neutral boundary-layer case would be insightful. Therefore, we performed an additional series of four single wind turbine and four wind farm simulations in truly neutral conditions. As now described in Sec.2.5, the TNBL is artificially obtained from the precursor of the case $H500$-$\Delta\theta5$-$\Gamma4$. The abstract and the results section (Sec. 4) is modified accordingly.

- ...wind farm power and thrust coefficient curves under three different conventionally neutral boundary-layer *and one truly neutral boundary-layer. As a result of the large-scale effects, we show that the wind farm power and thrust coefficient curves significantly deviate from those of*

*an isolated turbine. We carry out a trade-off analysis and determine that, while the optimal thrust set-point is still correctly predicted by the Betz limit under wake-only conditions, it shifts towards lower operating regimes under strong blockage conditions. In such cases, we observe a minor power increase with respect to the Betz thrust-set point, accompanied by a load reduction of about 5%.* More interestingly, we show that for some conditions the loads can be reduced by up to 19%, at the expense of a power decrease of only 1%.

- Note that for all three sets of atmospheric conditions, the capping inversion thickness is initialized to $\Delta \mathrm{H} = 100\,\mathrm{m}$. *In addition to the three CNBL atmospheric conditions, we consider a situation with no thermal stratification, similar to Lanzilao & Meyers (2024). To generate this flow, we start from the case H500-$\Delta\theta$5-$\Gamma$4 and artificially set a constant potential temperature profile when copying the solution from the precursor to the main domain. The resulting flow, denoted H500-$\Delta\theta$0-$\Gamma$0, resembles a truly neutral boundary-layer (TNBL), but with the same inlet velocity as H500-$\Delta\theta$5-$\Gamma$4.* Moreover, for all the simulations performed in this analysis, we set the geostrophic wind speed to...

- ...The same pattern can be seen, though to a lesser extent, in the case H300-$\Delta\theta$5-$\Gamma$1 Fig. 4 (e–h). *Under the set of conditions H500-$\Delta\theta$5-$\Gamma$4 (Fig. 4 (i–l)), large-scale effects are minor but become apparent when compared to the case H500-$\Delta\theta$0-$\Gamma$0 (Fig. 4 (m–p)). In particular, analyzing Fig. 4 (l) and Fig. 4 (p) together, we observe a slight velocity decrease limited to the front of the farm in Fig. 4 (l). Moreover, Fig. 4 (p) shows a stronger farm wake compared to Fig. 4 (l), providing evidence that a favorable pressure gradient still forms in situations where the capping inversion is high.* Eventually, observations of the vertical velocity field provided further evidence of the development of wind-farm-induced effects (not shown)...

- ...When operating at $C_T' = 0.5$ (Fig. 5 (a,e,i,m)), wake interference between the turbines dominates, which results in a region of higher thrust values over the first two rows of turbines, followed by a quasi-uniform distribution across the rest of the farm. *In the absence of a capping-inversion, the same conclusion applies regardless of the considered $C_T'$ value (Fig. 5 (m,n,o,p)).* For the CNBL conditions, the bow-wave pattern described in Fig. 4...

- ...This phenomenon appears to be clearly amplified for inflows with a low capping inversion (H150-$\Delta\theta$8-$\Gamma$1). *On the contrary, the non-local efficiency remains constant with respect to $C_T'$ in the absence of a capping inversion (case H500-$\Delta\theta$0-$\Gamma$0).* Under CNBL conditions, we notice in Fig. 8 (b)...

- ...followed by a favorable pressure gradient that accelerates the flow deeper into the farm, as previously visualized in Fig. 4 *Under the atmospheric conditions H500-$\Delta\theta$5-$\Gamma$4, the favorable pressure gradient leads to values of the farm efficiency larger than in the corresponding TNBL case (H500-$\Delta\theta$0-$\Gamma$0), as indicated in Fig. 8 (c). Similar observations are reported in Lanzilao & Meyers (2024).* Below $C_T' \simeq 1.25$, the farm poses so little resistance to the flow that only minor blockage effects occur...

**Reviewer Point P 1.11** — p17: "axial-induction control approaches". Choosing arbitrarily three different possible operating points and comparing the overall performances cannot be considered really a "control". I suggest modifying this part of the discussion, as well as the title of the paper

and the abstract/conclusion, referring rather to an "operational strategy" (or similar) instead of a control.

**Reply**: We agree with the reviewer on that. The title of the paper has now been changed to "A large-eddy simulation analysis of collective wind farm axial-induction set points in the presence of blockage". The term "control" has been changed as suggested in the manuscript.

**Reviewer 2**

This paper reports LES of a 10 x 10 staggered wind farm subjected to three different atmospheric conditions with the turbines operating at four different thrust coefficients. The aim is to investigate the effect of the axial induction factor (equivalent to changing each turbine's local thrust coefficient, CT') on the thrust and power coefficients of the whole wind farm. Using the LES data, empirical fits are provided for these quantities as a function of CT'. Using these fits, the authors conclude that allowing for a small reduction in the generated power can lead to a large reduction in the thrust experienced by the wind farm. The extent to which the CT and CP of the wind farm change on changing CT' is quantified for the different atmospheric conditions studied here. The paper is well-written and easy to read. I have three major comments and a few minor ones given below.

**Major comments**

**Reviewer Point P 2.1** — Section 4.4, Fig. 9, Tables 5, 6, 7: This is the main result of this paper. It is already known in the context of an isolated turbine (or implied by the inviscid momentum theory). It would be helpful to show the same result for an isolated turbine and comment on how things are different for a wind farm. Also, the effect of atmospheric conditions (inversion height, lapse rate and boundary-layer height) on the trade-off between CP,f, and CT,f should be studied for a wider range of atmospheric conditions to further support this main result. The authors already have a similar database (Lanzilao and Meyers, 2024) of wind farms subjected to a much wider range of atmospheric conditions which could be used for this analysis.

**Reply**: We thank the reviewer for this suggestion. We performed four wind farm simulations and four single turbine simulations under truly neutral boundary-layer conditions to enrich the discussion. Sections 2.5, 3 and 4 have been modified accordingly (also see Reviewer 1, point 1.10). Section 4.4, Fig. 9 and Tables 5,6 and 7 have been updated. In particular, Fig. 9 is now divided into 4 subfigures, namely, the results of the wind farm simulations and their normalized version, and the single turbine counterparts. The discussion in Sec. 4.4 has been re-written as below. We thank the reviewer for suggesting the use of the database of Lanzilao & Meyers (2024). However, it was generated with turbines of different dimensions over a different farm layout. Therefore, we took the decision to limit the study to the new LES results only.

[revised manuscript text omitted]

**Reviewer Point P 2.2** — Line 314: A least squares method is used to determine fitting coefficients. However, there are only 4 data points per atmospheric condition. How can 3 parameters be fit with only 4 data points? The same comment applies to Eq. (13) where there are further additional parameters but again only 4 data points. Line 315: The authors should clarify what they mean by 'thorough analysis' either in the main paper or in an appendix.

**Reply**: For the single turbine cases, the value of the three parameters is set through a joint least squares method over a total of eight points, i.e. 4 LES $C_T$ values and 4 LES $C_P$ values. This is no longer the

case for the wind farm operating curves, where the fitting parameters are set separately for $C_{T,f}$ and $C_{P,f}$. Therefore, a first least square fit is performed over the 4 LES $C_{T,f}$ points to set the values of the three parameters $\alpha_{t,f}$, $\delta_{t,f}$ and $\gamma_{t,f}$. Note that this is still a least squares fit, although there is only one measurement point more than needed for a perfect polynomial fit. The same procedure is carried out independently over the 4 LES $C_{P,f}$ values to determine the fitted values of $\alpha_{p,f}$, $\delta_{p,f}$ and $\gamma_{p,f}$. This is now discussed in the manuscript as follows:

- ...are fitted to the LES data points using the least squares method. *As the two relations in Eq. 12 share a common parameter $\beta$, the procedure results in a simple joint-optimization problem between the eight LES data points corresponding to the four tested $C_T'$ values. The optimized value of each of the three parameters is given in Table 3. In Eq. 12, an increasing number of parameters was introduced and a convergence analysis on the residual of the least squares method then motivated the choice of $\alpha_t$, $\alpha_p$ and $\beta$. Physically, we postulate that* the three parameters allow to account for the impact of shear, veer and turbulence, disregarded in the classical AMT...

- ...where six degrees of freedom are introduced in total. *In Eq. 13, the two sets of three fitting parameters are determined for $C_{T,f}$ and $C_{P,f}$ through two independent least squares methods. Each of the two fitting therefore sets the values of three parameters using four LES points. The corresponding values are tabulated in Table 4. We initially explored other options, e.g. using only three parameters to approximate $C_{T,f}$ and $C_{P,f}$ as in the single turbine case. However, this led to large fitting errors in all the tested cases.* The wind farm thrust and power coefficient curves shown in Fig. 7 can be discussed...

**Reviewer Point P 2.3** — In my understanding, the term 'control' is usually used in a dynamical sense, with different axial induction control strategies implying changing the CT' in response to flow conditions. In this paper, however, the thrust coefficient of each turbine is the same across the wind farm and is also frozen in a given simulation. Perhaps a title and phrasing throughout the paper such as 'sensitivity' to operating thrust coefficient would be more appropriate.

**Reply**: Thank you for your comment. We agree with you and we have now modified replaced the term "control" in the manuscript by "operational", as also suggested by Reviewer 1 (P1.11).

**Minor comments**

**Reviewer Point P 2.4** — What material in Section 2 is novel? For example, the wind turbine representation (Section 2.2) is standard unless I am missing some detail. Are the values mentioned in Tables 1 and 2 different from those used in previous work, i.e. Lanzilao and Meyers (2024)?

**Reply**: The methodology is indeed identical to that of Lanzilao & Meyers (2024). However, we note some minor differences, e.g., different turbine type and farm layout, larger simulation domain, different definition of the Shapiro's correction factor. The main novelty lies in the extensive range of operating conditions examined in this study, which, to the best of our knowledge, have not yet been investigated using LES.

**Reviewer Point P 2.5** — Figure 3: I do not see the black dashed line corresponding to the standard deviation.

**Reply**: Thanks for pointing this out. More details are now provided in the caption to make the figure more readable.

**Reviewer Point P 2.6** — The contours in Fig. 4 appear pixellated and patchy (small rectangles). Is this because of the LES resolution or a plotting artefact? Given the size of the rectangles relative to the turbine diameter, it is probably the way each contour plot is exported and not the LES resolution.

**Reply**: The results of the truly neutral boundary-layer have been added to the figure and the resolution has been improved.

**Reviewer Point P 2.7** — Sentence on lines 247-249: "This can be visualized by . . . straight in the latter case": This is not very clear. Drawing some lines that visualize the wake extents on these contour plots might help, or this should be shown using some profiles. Also, it is not clear how a faster wake recovery is associated with the horizontal extent of the wake remaining 'straight'.

**Reply**: We thank the reviewer for this point. We agree that this cannot clearly be seen on the figure. As the evolution of farm wakes is slightly out of the scope of this work, we left out this paragraph. However, the flow under truly neutral boundary-layer conditions is now discussed in that section.

**Reviewer Point P 2.8** — Line 262: Why are chord lengths considered in calculating the reference speed? The turbine model is a thrusting actuator disk where the chord length is not an input. Why not simply use a disk-average?

**Reply**: Thank you for pointing out this lack of clarity. The domain over which the reference speed is computed is $[0, L_x^p] \times [0, L_y^p] \times [z_H - D/2, z_H + D/2]$ in the precursor field. The weights used for the weighted average along the vertical directions are given by the disk chord length, i.e. the straight-line distance across the intersection of the disk and the horizontal plane at the considered height. This is now clarified in the manuscript as follows:

Further, $U_\infty$ is the reference wind speed computed as the streamwise velocity averaged over a layer of thickness $D$ spanning the disc-precursor domain, *i.e. the region defined by* $[0, L_x^p] \times [0, L_y^p] \times [z_H - D/2, z_H + D/2]$. *Within this region, we use a vertically dependent weighted average where the weights are given by the actuator disc chord length, i.e., the straight-line distance across the intersection of the disc and the horizontal plane at the considered altitude.* For the cases H150-$\Delta\theta$8-$\Gamma$1, H300-$\Delta$5-$\Gamma$1, H500-$\Delta$5-$\Gamma$4 and H500-$\Delta\theta$0-$\Gamma$0, the reference speeds...

**Reviewer Point P 2.9** — Line 266: Please clarify in the text that 'disk-based coefficient' means CT.

**Reply**: This is now explicitly specified multiple times in the manuscript to avoid ambiguities.

**Reviewer Point P 2.10** — Line 272: Do the authors mean Fig. 5(d) here? I cannot distinguish colours between the first few rows in Fig. 5 (d). I understand the overall point that the colour

differences are more drastic for panels (a, e, i). But if the authors are commenting on differences between the rows in Fig. 5(d), it would be better to use a different colour scheme where these differences show up more clearly.

**Reply**: We agree with the reviewer and we have improved the readability of the figure in two ways. First, the plot now represents the thrust distribution normalized by the thrust value in the corresponding isolated case. As discussed in the manuscript (see below), this is equivalently expressed in terms of the ratio of the local thrust coefficient and the single turbine thrust coefficient: $C_{T,k}/C_{T,sgl}$. This allows to significantly enhance the contrast between the turbines within the farm. Second, the farm is now represented with an hexagonal paving to remove the white spacing between the turbines and improve clarity.

We note that the two expressions in Eq. 8 can be re-written as $C_{T,k} = C_T'(\overline{u}_{d,k}/\overline{U}_\infty)^2$ and $C_{P,k} = C_T'(\overline{u}_{d,k}/\overline{U}_\infty)^3$ using Eq. 5 and Eq. 7, with $\overline{u}_{d,k}$ the time average of $u_{d,k}$. *The time-averaged thrust ($C_{T,sgl}$) and power ($C_{P,sgl}$) coefficients in the single turbine case are defined with respect to the corresponding thrust ($\overline{F}_{sgl}$) and power ($\overline{P}_{sgl}$), analogously to Eq. 8. The distribution of the local thrust coefficient ($C_{T,k}$) over the farm is normalized by that of the single turbine ($C_{T,sgl}$) under the same operating conditions, and represented in Fig. 5. Therefore, Fig. 5 illustrates the momentum extracted by each turbine in the farm, compared to that of an isolated turbine. Because the disc-based thrust coefficient is common to each turbine in the farm, the ratio shown in Fig. 5 re-writes $C_{T,k}/C_{T,sgl} = \overline{F}_k/\overline{F}_{sgl} = (\overline{u}_{d,k}/\overline{u}_{d,sgl})^2$. When operating at $C_T' = 0.5$ (Fig. 5 (a,e,i,m)), wake interference between the turbines dominates, which results in a region of higher thrust values*

**Reviewer Point P 2.11** — Eq. (13) is missing a 'CT = '. Also, there should be a subscript 'f' on these if these are farm-averaged quantities.

**Reply**: This has been changed in the manuscript.

**Reviewer Point P 2.12** — Some comments on how sensitive these findings would be to the wind turbine type (e.g. diameter, hub-height) and surface roughness values would be appreciated by the readers.

**Reply**: We thank the reviewer for this very relevant suggestion. We propose four non-dimensional groups that, we believe, capture most of the flow similarities. This is addressed in the text as:

...We believe this constitutes an important finding, upon which more sophisticated wind farm operational strategies can be developed. *In the future, investigating the sensitivity of the results to the turbine type, the farm layout or the free-stream velocity could be of interest. Regarding the ABL flow profile, we anticipate that the diameter-to-hub-height ratio and the ratio of the roughness length to the hub height are meaningful to the problem. We denote them $D^* = D/z_H$ and $z_0^* = z_0/z_H$, respectively. Further, we follow the expression of the similarity parameter $H^* = |f_c|H/u_*$ (Sood, 2023), where $f_c$ is the Coriolis frequency, $H$ is the boundary-layer height and $u_*$ is the friction velocity. We note that the hub height velocity $U_H$ can substitute $u_*$, using a log-law profile and the parameter $z_0^*$ defined above. Throughout the present work, $D^*$ and $z_0^*$ were kept constant whereas*

*different values of $H^*$ were considered. We refer to, e.g. Csanady et al. 1974, to relate $H^*$ to the potential temperature parameters set in Sec. 2.5. Lastly, the present work provided evidence of the substantial impact of $C_T'$ on the flow dynamics. More generally, we expect power density to play a crucial part in the design of an effective farm operating strategy. Therefore, we introduce the disc-based friction coefficient factor defined in Calaf et al.(2010) as the fourth non-dimensional number to account for power density. This ratio reads $c_{ft}' = \pi C_T'/(4 S_x S_y)$, where $S_x$ and $S_y$ are the turbine spacings (expressed in number of diameters) in the streamwise and spanwise directions, respectively. As a result, similar effects on the total power extraction may be expected for similar values of $c_{ft}'$.*